# Temporal Reasoning Transfer from Text to Video

**Lei Li**[1*]  **Yuanxin Liu**[2*]  **Linli Yao**[2]  **Peiyuan Zhang**[3]  **Chenxin An**[1]
**Lean Wang**[2]  **Xu Sun**[2]  **Lingpeng Kong**[1]  **Qi Liu**[1]
[1]The University of Hong Kong  [2]Peking University  [3]University of California, San Diego
nlp.lilei@gmail.com  {liuyuanxin, linliyao}@stu.pku.edu.cn
pez010@ucsd.edu  cxan23@connect.hku.hk
{lean, xusun}@pku.edu.cn  {lpk,liuqi}@cs.hku.hk

## Abstract

Video Large Language Models (Video LLMs) have shown promising capabilities in video comprehension, yet they struggle with tracking temporal changes and reasoning about temporal relationships. While previous research attributed this limitation to the ineffective temporal encoding of visual inputs, our diagnostic study reveals that video representations contain sufficient information for even small probing classifiers to achieve perfect accuracy. Surprisingly, we find that the key bottleneck in Video LLMs' temporal reasoning capability stems from the underlying LLM's inherent difficulty with temporal concepts, as evidenced by poor performance on textual temporal question-answering tasks. Building on this discovery, we introduce the **T**extual **T**emporal reasoning **T**ransfer (**T3**). T3 synthesizes diverse temporal reasoning tasks in pure text format from existing image-text datasets, addressing the scarcity of video samples with complex temporal scenarios. Remarkably, *without using any video data*, T3 enhances LongVA-7B's temporal understanding, yielding a 5.3 absolute accuracy improvement on the challenging TempCompass benchmark, which enables our model to outperform ShareGPT4Video-8B trained on 28,000 video samples. Additionally, the enhanced LongVA-7B model achieves competitive performance on comprehensive video benchmarks. For example, it achieves a 49.7 accuracy on the Temporal Reasoning task of Video-MME, surpassing powerful large-scale models such as InternVL-Chat-V1.5-20B and VILA1.5-40B. Further analysis reveals a strong correlation between textual and video temporal task performance, validating the efficacy of transferring temporal reasoning abilities from text to video domains.[1]

## 1 Introduction

The rapid development of large language models (LLMs) (OpenAI, 2024; Gemini Team, 2024) has sparked significant interest in video large language models (Video LLMs) (Zhang et al., 2023; Lin et al., 2023b) due to their impressive generation and reasoning capabilities. Current approaches typically use pre-trained vision encoders (Radford et al., 2021) combined with powerful LLMs (Touvron et al., 2023; Chiang et al., 2023; Yang et al., 2024) as the starting point for Video LLMs. These models employ various strategies to handle multiple video frames (Li et al., 2023b; Tan et al., 2024), and are then further trained on curated instruction-tuning datasets (Chen et al., 2024), demonstrating promising abilities in video comprehension tasks (Fu et al., 2024; Zhou et al., 2024).

Despite the progress in video comprehension, Video LLMs often struggle with temporal reasoning, which is essential for truly interpreting video content (Li et al., 2023c; Tang et al., 2023b). Specifically, video temporal reasoning involves the ability to track changes over time, comprehend event sequences, and relate objects and actions to specific moments in a video (Mangalam et al., 2023; Li et al., 2023d). As illustrated in Figure 1, two strong Video LLMs, LongVA-7B (Zhang et al., 2024b) and VILA-8B (Lin et al., 2023b), both failed to answer basic questions about the chronological order of events

---

[*]Equal contribution.
[1]Project page: https://video-t3.github.io

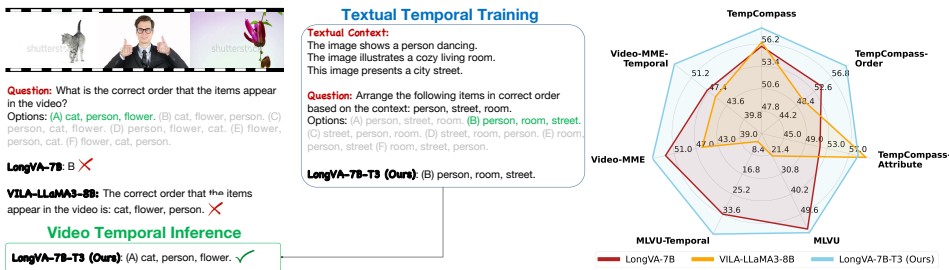

Figure 1: Two popular Video LLMs struggle with basic temporal reasoning (**left**). We mitigate this issue via textual temporal transfer (**middle**), which demonstrates consistent improvement (**right**).

in synthesized videos, whereas humans can predict correctly without difficulty. This significant gap between human performance and current Video LLMs in temporal reasoning tasks motivates us to explore the underlying reasons for this discrepancy.

Previous research has largely attributed temporal reasoning deficiencies in Video LLMs to ineffective video encodings, leading to various temporal aggregation module developments (Ren et al., 2023; Jin et al., 2023; Tan et al., 2024). Our paper takes a different approach by decomposing Video LLMs into two parts and asking a fundamental question: What is the bottleneck of this limitation? Is it due to limitations in the vision encoder, or, surprisingly, shortcomings in the LLM itself? We conduct probing experiments using synthesized videos for basic temporal-related video question-answering (QA) tasks, allowing full control over temporal aspects. Our method involves: (i) Training small probe classifiers on video representations in the Video LLM embedding space to assess temporal information captured by visual encoders and aggregation modules. (ii) Transforming synthesized videos into textual descriptions using commercial visual language models (e.g., GPT-4o) to analyze how standalone LLMs process temporal information. By comparing the performance of these components with full Video LLMs, we could precisely locate the bottleneck in temporal understanding.

Our experiments reveal a striking contrast in temporal reasoning capabilities between different components of Video LLMs. Probe classifiers trained on video embeddings achieve near-perfect accuracy ($> 90\%$ in most cases), indicating that these embeddings successfully capture sufficient temporal information. Even simple neural models like LSTM (Hochreiter & Schmidhuber, 1997) can accurately extract temporal relationships from these embeddings. Conversely, despite their significantly larger scale, LLMs struggle to process this temporal information effectively. They exhibit relatively low probing accuracy across various temporal aspects, highlighting their difficulty in handling temporal relationships. These findings provide compelling evidence that **the LLM component, rather than the visual encoding, is the primary bottleneck in the temporal reasoning of current Video LLMs**, motivating us to enhance LLMs' ability to reason about temporal information.

Building on our insights, we propose enhancing Video LLMs' temporal understanding by focusing on the LLM component. Our approach leverages existing image-text datasets to generate diverse temporal reasoning tasks in *pure text format*, overcoming the scarcity of video samples with complex temporal scenarios (§3). Remarkably, without using any video data, our text-only synthesized dataset enables LongVA-7B to outperform ShareGPT4Video-8B (trained on 28,000 video samples) on the TempCompass benchmark (Liu et al., 2024b). Moreover, as shown in the right of Figure 1, our enhanced model demonstrates competitive results on various video-understanding benchmarks. It improves temporal reasoning accuracy by 12.4 points on Video-MME (Fu et al., 2024) and increases average accuracy from 56.4 to 58.1 on MLVU (Zhou et al., 2024), surpassing larger models such as InternVL-Chat-V1.5-20B (Chen et al., 2023) and VILA-1.5-40B (Lin et al., 2023b). Analysis reveals the crucial role of self-attention modules in temporal reasoning transfer, and our method improves the utilization of more video frames and transfers to subtle temporal aspects well (Shangguan et al., 2024; Zhang et al., 2024a).

## 2    PINPOINTING VIDEO LLM TEMPORAL REASONING BOTTLENECK

In this section, we seek to examine and pinpoint the temporal reasoning bottleneck of Video LLMs. Video LLMs typically consist of two essential components for temporal understanding tasks: a vision

encoder and an LLM decoder, where the former extracts visual features from video frames and the latter is responsible for integrating this information with textual instructions to complete the end task. We aim to answer two questions: (1) Can existing Video LLMs understand the temporal information in videos? (2) If not, which component—the vision encoder or the LLM decoder—is the bottleneck? To address these questions, we design different tasks to test the full Video LLMs and these two components separately (§2.1), incorporating various aspects of temporal understanding abilities (§2.2), and discuss our findings (§2.3).

## 2.1 TASK FORMULATION FOR DIFFERENT VIDEO LLM COMPONENTS

**Full Video LLM.** We evaluate the full Video LLMs through multi-choice video question answering. Specifically, we uniformly sample eight frames from videos and present them to the model along with a multiple-choice question. To encourage the Video LLM to directly output an option, we append the prompt "*Answer the option only.*" after the question.

**LLM Decoder.** We replace the video frames with detailed frame captions generated by GPT-4o. In this way, we assess the ability of LLM decoder to understand temporal information in the textual context. The multi-choice questions are identical to those employed in testing the full Video LLM. To ensure that the temporal understanding questions can indeed be addressed using these frame captions, we carefully design the prompting strategy to incorporate all essential information in the captions (please refer to Appendix A.2 for the implementation details and examples of frame captions.)

**Visual Features.** Unlike the full Video LLM and LLM decoder, the quality of visual features cannot be directly tested via question answering. To determine whether the visual features contain sufficient information to differentiate between different temporal dynamics in videos (e.g., *brightening* versus *darkening*), we employ the "classifier probe" technique proposed by (Alain & Bengio, 2017). Assuming that there are $c$ categories of contradicting temporal dynamics, we train a "probe" $f()$ that maps a set of visual features $\mathbf{V} \in \mathcal{R}^{n \times d_v}$ to a probability distribution $p \in \mathcal{R}^c$. In practice, $f()$ is a single-layer LSTM model (Hochreiter & Schmidhuber, 1997) for capturing sequential correlation, and we employ the visual features that are down-sampled and projected into LLM embedding space. The probe is tested on the same set of videos as the full Video LLM but is trained on a different set of videos. More details of the visual classifier probe can be found in Appendix A.3.

## 2.2 EVALUATION DATA COLLECTION

**Temporal Reasoning Abilities.** Temporal reasoning encompasses several key aspects. Based on recent Video LLM benchmarks (Fu et al., 2024; Wu et al., 2024), we focus on four critical dimensions: (1) **Order**: comprehending the sequential arrangement of events; (2) **Attribute**: perceiving changes in environmental or object attributes over time; (3) **Temporal Referring**: formulating questions based on specific temporal positions within a video; and (4) **Temporal Grounding**: identifying the temporal location of specific elements in a video. While these aspects are a limited approximation of the full spectrum of temporal understanding, our study reveals that they are sufficient to expose significant deficiencies in current Video LLMs.

**Collecting Videos.** Existing video benchmarks are unsuitable for our analysis due to two main limitations: (1) the lack of video clusters with contrasting temporal dynamics needed for visual feature probing, and (2) the inability to fully eliminate single-frame or language bias (Liu et al., 2024b). Consequently, we synthesize custom videos to isolate different aspects of temporal understanding. Figure 2 illustrates examples of our synthesized videos. For the *Order* aspect, we concatenate different source videos (e.g., person, cat, flower) temporally, creating various categories based on concatenation order. The *Attribute* aspect focuses on shape (flower blooming videos and their reversals) and brightness (gradually altering pixel values of static images). *Temporal Referring* and *Temporal Grounding* reuse videos from the Order aspect with three concatenated items. Appendix A provides detailed information on the video creation process and data distribution.

**Constructing Questions and Answers.** As depicted in Figure 2, we design multi-choice question templates for each temporal aspect. The questions remain consistent across videos within each aspect, while the correct answers vary based on the specific video content. This approach ensures a controlled evaluation of temporal reasoning capabilities.

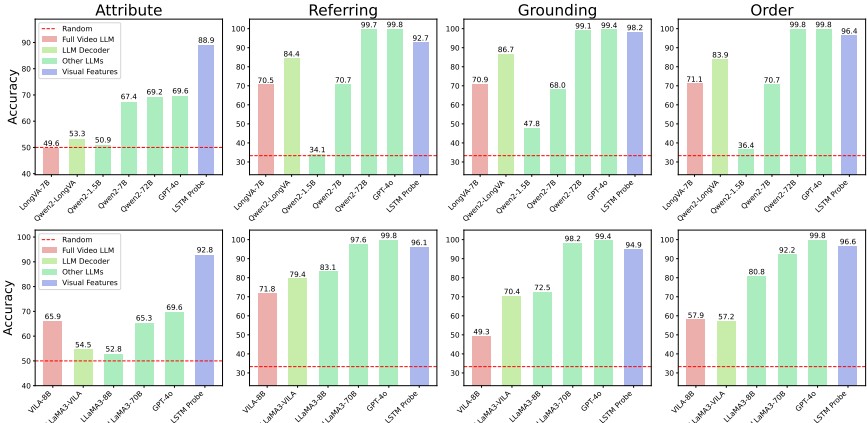

Figure 2: Example of videos and questions focusing on different temporal reasoning abilities.

Figure 3: Temporal probing results for LongVA (upper) and VILA (lower). The probe on visual representations achieves > 90 accuracy in most cases, while the LLM decoders still have large room for improvement even with textual inputs, leading to the poor temporal understanding ability of Video LLMs. Detailed results of the sub-categories are reported in Appendix A.4.

## 2.3 RESULTS

We examine two advanced Video LLMs that utilize different LLM backbones on our synthesized probing datasets: LongVA-7B (Zhang et al., 2024b), which is based on Qwen2-7B (Yang et al., 2024), and VILA-8B (Lin et al., 2023b), which is built upon LLaMA3-8B (Dubey et al., 2024). The results are visualized in Figure 3, with more results reported in Appendix A.4. Our analysis reveals several key findings:

**Performance of Video LLMs:** Video LLMs demonstrate relatively poor performance, struggling to reach 70 accuracy across all temporal understanding tasks. This suggests a potential limitation in their ability to process and reason about temporal information in video content.

**Efficacy of Visual Representations:** Notably, small classifiers trained on the visual representations in the LLMs' embedding space achieve near-perfect accuracy. This finding suggests that these visual representations encapsulate rich information, sufficient to distinguish between temporally contradicting videos, even with a simple probe classifier. Consequently, we can conclude that **the input processing is not the primary limitation in temporal reasoning tasks.**

**LLM Backbone Performance:** To our surprise, moderate-sized LLM backbone decoders such as Qwen2-7B and LLaMa3-8B fail to answer a considerable number of questions related to *Referring*, *Grounding* and *Order* aspects, which are intuitively very easy for SoTA LLMs given that the frame captions are provided in textual format. Worse still, these LLM decoders perform only at the level of random guessing when it comes to the *Attribute* aspect. These findings indicate that **the LLM decoders face substantial challenges even in the context of textual temporal understanding, posing a major limitation to the video temporal understanding capabilities of Video LLMs**. In comparison, larger LLMs with over 70 billion parameters significantly outperform their smaller counterparts on these tasks, suggesting that (i) the generated frame captions are accurate and contain

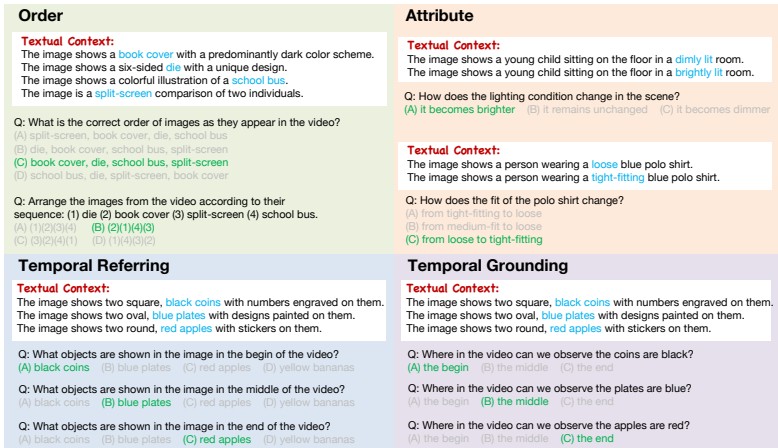

Figure 4: Temporal-oriented question-answering pairs with textual image captions as context.

sufficient information for temporal reasoning; (ii) the textual temporal understanding ability emerges when the text decoder scale exceeds certain thresholds. We hypothesize that this discrepancy may be due to the sparse expression of temporal concepts in pre-trained corpora. Consequently, smaller-scale models might not have sufficient exposure to learn these temporal reasoning abilities effectively. This sparsity could explain why only very large language models (>70B parameters) demonstrate proficiency in temporal reasoning tasks.

Our findings ultimately suggest that the temporal reasoning ability in current Video LLMs is primarily constrained by the LLM decoder rather than the quality of visual representations. This conclusion highlights a critical area for improvement in the development of more effective Video LLMs, motivating us to enhance the temporal reasoning capabilities from the textual side.

## 3 TEXTUAL TEMPORAL UNDERSTANDING TRANSFER

In this section, we explore strategies to mitigate the deficiency of Video LLMs in temporal reasoning. A straightforward approach would be to create a temporal-oriented instruction-tuning dataset from videos. However, existing approaches to video instruction-tuning data generation, whether through human annotation or automatic generation via rules/models, present significant challenges. Human annotation of video datasets is resource-intensive, demanding considerable time and financial investment. Synthetic video instruction tuning (Maaz et al., 2024; Zhang et al., 2024c), while more efficient and scalable, is constrained by the content of the videos themselves. For instance, if our goal is to enhance a particular aspect of temporal reasoning, it is crucial to ensure that the videos contain information relevant to this aspect. This limitation makes it difficult to create instruction-tuning data that targets specific abilities.

Motivated by (1) the limitations of video instruction-tuning and (2) our finding that the LLM backbone is the primary bottleneck of temporal understanding, we investigate the feasibility of enhancing video temporal reasoning through a novel perspective: using synthesized textual temporal reasoning data. Our method uses sequences of image captions as proxies for video frames, allowing us to create temporal-oriented question-answering pairs without relying on actual video content. This text-only approach offers two key advantages: (i) **Scalability**: The abundance of available image-caption pairs (Changpinyo et al., 2021; Liu et al., 2023) allows for easy expansion of the dataset. (ii) **Flexibility**: We can precisely control the targeted ability and sample complexity by adjusting the content and number of image captions. To address the four aspects of temporal understanding (*Order*, *Attribute*, *Referring* and *Grounding*), we design heuristics to construct textual contexts and generate question-answer pairs solely from caption sequences with examples visualized in Figure 4. The specific generation processes are as follows.

**Order.** This aspect focuses on understanding the sequential order of image captions. To construct the textual context for question answering, we randomly sample 3∼6 captions from a caption pool.

Table 1: Summary of our textual temporal reasoning datasets where $\mathbf{X} \in \{\text{phrase, prefix, sentence}\}$ for `Order-Template` ($\mathbf{X}$). Detailed data statistics are provided in Appendices B.3 and C.1.

| Dataset | #Relevant Captions | #Distractor Captions | Description |
|---|---|---|---|
| Order-GPT ($N\times$) | 2~4 | $N \times 100 \pm 50, N \in \{1, 2, 4, 8\}$ | Order-related questions generated by GPT-4. |
| Attribute ($N\times$) | 2 | $N \times 100 \pm 50, N \in \{1, 2, 4, 8\}$ | Attribute-related questions. |
| Order-Template ($\mathbf{X}$) | 3~6 | $200\pm50$ | Order-related questions based on templates $\mathbf{X}$ |
| Referring | 3 | $200\pm50$ | Temporal referring questions. |
| Grounding | 3 | $200\pm50$ | Temporal grounding questions. |

The questions are created using two methods: On the one hand, we provide GPT-4-turbo with examples and prompt it to generate order-related questions, which we refer to as **Order-GPT**. On the other hand, we employ predefined templates and heuristic rules to create questions. These template questions require rearranging shuffled sequences according to the textual context of image captions. These sequences comprise of (a) complete image captions, (b) phrases within captions or (c) phrases with prefix identifiers (e.g., *(1)(2)(3)(4)* as shown in the example in Figure 4). We denote the template-based questions as **Order-Template (X)**, where X $\in \{\text{sentence, phrase, prefix}\}$.

**Attribute.** This aspect involves recognizing how specific attributes of objects or scenes change throughout the video. To achieve this, we first prompt GPT-4-turbo to generate new image captions by modifying particular attributes in the original captions. Specifically, we consider five types of attributes: *color*, *light condition*, *size & shape*, *posture*, and *emotion*. Subsequently, we employ GPT-4-turbo again to generate questions that focus on the attribute changes between pairs of captions.

**Temporal Referring.** This aspect requires understanding questions that refer to particular temporal locations (e.g., *begin*, *middle* and *end*). To increase the difficulty, we employ GPT-4-turbo to generate three similar image captions that only differ in certain aspect, such as object, action or attribute. These three captions are then placed respectively at the beginning, middle, and end of the textual context. For the questions, we first generate a question for each caption without referring to temporal locations. Then, temporal references are added to the questions according to the specific temporal location of the corresponding caption.

**Temporal Grounding.** This aspect is a complementary angle to *temporal referring*, aiming to identify the temporal location of an element (a phrase describing an object, action or attribute) of interest. The textual context reuses the same image captions as in *temporal referring*. To formulate the questions, we first prompt GPT-4-turbo to generate a declarative statement for each caption (e.g., "the coins are black"). These statements are then incorporated into the temporal grounding question templates, as illustrated in Figure 4.

We enhance the basic textual context (i.e., captions relevant to the question) by incorporating "distractor captions" sampled from the caption pool and inserted between the original captions. This approach challenges the model to identify relevant temporal information amidst irrelevant context, enhances the robustness of its temporal understanding abilities, and presents a more realistic scenario mimicking the complexity of real-world temporal reasoning tasks. To maintain question clarity, we ensure distracting captions do not share nouns with the original captions. Table 1 gives a summary of our synthesized textual temporal QA datasets. For each task, we also create a validation set consisting of 500 samples for later verification. Appendix B provides a detailed and formalized description of the data construction process, as well as the ablation study for distractor captions and a comparison of temporal reasoning transfer via multiple images (Appendix B.4).

## 4 EXPERIMENTS

### 4.1 EXPERIMENTAL SETTINGS

**Benchmarks.** To establish a connection between textual temporal reasoning ability and video comprehension, we adopt the fine-grained temporal understanding benchmark, TempCompass (Liu et al., 2024b) (Multiple-Choice subset), diagnosing multi-facet basic video temporal understanding abilities. For a comprehensive assessment of long-form video understanding capabilities, we choose two challenging benchmarks: MLVU (Zhou et al., 2024) and Video-MME (Fu et al., 2024). These benchmarks are specifically designed to test temporal reasoning across diverse domains. For MLVU,

Table 2: TempCompass evaluation results. Without training on any videos, our T3 helps LongVA-7B outperform ShareGPT4Video-8B trained on 28k video samples. `Temporal Change` combines `Order-GPT` and `Attribute`. The best results are shown in **bold**.

| Method | Action | Direction | Speed | Order | Attribute Change | Average |
|---|---|---|---|---|---|---|
| Video-ChatGPT-7B (Maaz et al., 2024) | 61.5 | 28.9 | 29.0 | 36.1 | 30.9 | 37.7 |
| Video-LLaVA-7B (Lin et al., 2023a) | 76.0 | 35.2 | 35.7 | 37.8 | 41.0 | 45.6 |
| LLaVA-NeXT-Video-7B-DPO (Liu et al., 2024a) | 87.6 | 35.8 | 41.3 | 39.7 | 45.8 | 50.6 |
| Llama-3-VILA1.5-8B (Lin et al., 2023b) | 92.9 | 33.7 | 44.2 | 50.0 | 60.1 | 56.4 |
| ShareGPT4Video-8B (Chen et al., 2024) | 87.6 | 34.6 | **47.5** | 62.9 | 64.2 | 59.4 |
| LongVA-7B (32 frm) | **92.3** | 37.3 | 42.0 | 54.3 | 51.7 | 55.9 |
| + LLaVA-Next | 92.0 | 36.4 | 43.2 | 55.6 | 51.0 | 56.0 |
| + Hotpot QA | **92.3** | 37.0 | 39.4 | 53.0 | 50.7 | 54.9 |
| + Temporal Change (1x) | 91.1 | 37.3 | 39.8 | 65.2 | 63.9 | 59.5 |
| + Temporal Change (2x) | 89.6 | 39.1 | 40.4 | **66.9** | 61.8 | 59.6 |
| + Temporal Change (4x) | 90.5 | **43.0** | 39.4 | 64.2 | 64.2 | 60.4 |
| + Temporal Change (8x) | 90.2 | 42.4 | 39.8 | 64.2 | **68.1** | 61.0 |
| + Temporal Change (1x, 2x, 4x and 8x) | 90.5 | 38.2 | 39.8 | 65.6 | 64.2 | 59.7 |
| + Order-Template | 89.9 | 35.8 | 46.1 | 58.3 | 54.5 | 57.2 |
| + Temporal Referring | 91.4 | 33.7 | 46.1 | 46.7 | 51.0 | 54.2 |
| + Temporal Grounding | 63.0 | 31.6 | 29.3 | 40.7 | 38.9 | 41.0 |
| + Order-Template + Temporal Change (1x) | 90.2 | 40.6 | 42.9 | 63.6 | 65.3 | 60.6 |
| + Temporal Grounding+ Temporal Change (1x) | 91.1 | 38.8 | 41.6 | 62.9 | 62.9 | 59.6 |
| + Temporal Referring + Temporal Change (1x) | 90.8 | 38.2 | 42.3 | 62.9 | 64.6 | 59.8 |
| + T3 (all tasks) | 90.8 | 39.7 | 41.6 | 65.9 | **68.1** | **61.2** |
| GPT-4o | 98.2 | 52.8 | 52.1 | 73.2 | 78.5 | 71.0 |

Table 3: MLVU evaluation results. Our textual temporal reasoning transfer achieves the best overall performance, with significant gains in temporal-related aspects over the backbone model. TR: Topic Reasoning, AR: Anomaly Recognition, ER: Ego Reasoning, AO: Action Order, AC: Action Count. * denotes temporal-related dimensions. Best results are in **bold**.

| Model | AC* | ER | Needle QA | AO* | Plot QA | AR | TR | Macro Average |
|---|---|---|---|---|---|---|---|---|
| Video-ChatGPT-7B (Maaz et al., 2024) | 31.1 | 42.0 | 40.3 | 25.1 | 29.9 | 24.0 | 26.9 | 31.3 |
| Video-LLaVA-7B (Lin et al., 2023a) | 35.9 | 45.2 | 53.2 | 20.1 | 48.4 | 57.0 | 71.6 | 47.3 |
| MA-LMM-7B (He et al., 2024) | 24.3 | 38.9 | 43.1 | 25.1 | 35.8 | 35.5 | 51.9 | 36.4 |
| Llama-3-VILA1.5-8B (Lin et al., 2023b) | 0.0 | 24.7 | 32.4 | 6.6 | 20.0 | 27.0 | 46.2 | 22.4 |
| VILA1.5-40B (Lin et al., 2023b) | 11.7 | 35.8 | 38.3 | 34.3 | 62.0 | 56.4 | 84.7 | 46.2 |
| InternVL-Chat-V1.5-20B (Chen et al., 2023) | 13.3 | 24.5 | 40.0 | 14.3 | 42.0 | 51.3 | 80.2 | 37.9 |
| LongVA-7B (128 frm) | 25.2 | 48.6 | 70.4 | 41.7 | 68.1 | **58.5** | **82.2** | 56.4 |
| + LLaVA-Next | 11.7 | 20.5 | 36.6 | 17.8 | 45.3 | 17.5 | 72.7 | 31.7 |
| + Hotpot QA w/ LLaVA-Next | 13.1 | 27.6 | 40.6 | 19.7 | 45.3 | 21.5 | 72.4 | 34.3 |
| + T3 w/ LLaVA-Next (Ours) | **29.1** | **48.9** | **72.1** | **54.4** | **69.4** | 51.0 | 81.4 | **58.1** |
| GPT-4o | 46.3 | 57.1 | 64.8 | 56.7 | 65.1 | 74.5 | 87.4 | 64.6 |

we select the following tasks covering three aspects: holistic video understanding (Topic Reasoning (TR) and Anomaly Reasoning (AR)), single-detail understanding (Needle Question-Answering, Ego Reasoning (ER) and Plot Question Answering), and multi-detail understanding (Action order (AO) and Action Count (AC)). For Video-MME, which consists of 2700 video QA pairs spanning primary visual domains and three video duration types (Short, Medium, and Long), we focus on two temporal-oriented tasks (i.e., Temporal Perception and Temporal Reasoning) and overall performance.

**Compared Methods.** We evaluate our temporal textual augmentation approach against two baselines: (i) LLaVA-Next: Continually fine-tuning on the original image-text instruction dataset used by LongVA (Chen & Xing, 2024). (ii) HotpotQA (Yang et al., 2018): Fine-tuning on a multi-document QA dataset to assess enhancements in textual locating ability. We also report the performance of recent Video LLMs, including Video-ChatGPT (Maaz et al., 2024), Video-LLaVA-7B (Lin et al., 2023a), LLaVA-Next-Video-7B-DPO (Liu et al., 2024a), MA-LMM-7B (He et al., 2024), ShareGPT4Video-8B (Chen et al., 2024), LLama-3-VILA1.5-8B (Lin et al., 2023b), VILA1.5-40B (Lin et al., 2023b), and InternVL-CHat-V1.5-20B (Chen et al., 2023). GPT-4o (OpenAI, 2024) is included to represent commercial model results. These models serve as reference points and we report the highest scores published to represent optimal performance.

**Training Details.** We adopt LongVA (Zhang et al., 2024b) as our backbone model and continually fine-tune the official checkpoint. Despite being trained exclusively on text and image data, LongVA

Table 4: Video-MME evaluation results. Our method enhances LongVA-7B's accuracy across various video durations, even outperforming VILA1.5-40B in temporal reasoning. Best results are in bold.

| Model | Temporal Perception | Temporal Reasoning | Short | Medium | Long | Overall |
|---|---|---|---|---|---|---|
| Video-LLaVA-7B (Lin et al., 2023a) | - | - | 45.3 | 38.0 | 36.2 | 39.9 |
| LLaVA-NeXT-Video-7B-DPO (Liu et al., 2024a) | 40.0 | 29.4 | 48.9 | 42.0 | 35.6 | 42.1 |
| Llama-3-VILA1.5-8B (Lin et al., 2023b) | 50.9 | 41.2 | 56.1 | 42.1 | 39.6 | 45.9 |
| VILA1.5-40B (Lin et al., 2023b) | 60.0 | 40.7 | **72.0** | 61.2 | **53.8** | **62.3** |
| InternVL-Chat-V1.5-20B (Chen et al., 2023) | 45.5 | 33.3 | 60.2 | 46.4 | 45.6 | 50.7 |
| LongVA-7B (128 frm) | 58.2 | 37.3 | 61.1 | 50.4 | 46.2 | 52.6 |
| + LLaVA-Next | 54.6 | 37.3 | 61.2 | 50.6 | 44.9 | 52.2 |
| + Hotpot QA w/ LLaVA-Next | **65.5** | 39.6 | 60.2 | 50.9 | 45.6 | 52.2 |
| + T3 w/ LLaVA-Next (Ours) | 60.0 | **49.7** | 63.3 | 54.8 | 46.8 | 55.0 |
| GPT-4o | 74.1 | 59.4 | 80.0 | 70.3 | 65.3 | 71.9 |

demonstrates exceptional zero-shot video understanding capabilities and can effectively handle long videos. We strictly adhere to the original LongVA implementation and follow the official fine-tuning protocol. This involves using Adam (Kingma & Ba, 2015) as the optimizer, with learning rates of 2e-6 for the visual encoder and 1e-5 for the rest part of the model. For the exploration of textual temporal reasoning transfer, we use 22k samples for fine-tuning across different augmentation datasets. We further scale the dataset up to transfer to holistic video understanding benchmarks, where we find it necessary to incorporate the original instruction tuning data to maintain visual perception ability. According to our textual validation accuracy, we set the ratio of textual temporal QA and original data to 1:2 and the total samples to 200k. This mixing ratio applies to Hotpot QA for a fair comparison. Appendix C provides the details of dataset composition mixture configurations, and the dataset scaling effect. The model is trained on the corresponding dataset for one epoch, a process that can be completed within 5 hours using 8 H100 GPUs. We apply our recipe to a larger Qwen2-VL-72B backbone and observe consistent improvements for temporal reasoning (Appendix D.1).

## 4.2 RESULTS

Our results aim to answer the following three research questions: (1) Can textual temporal reasoning ability be effectively transferred to video temporal reasoning? and (2) Do these two abilities correlate well? and (3) Does this transfer also reflect on holistic video understanding benchmarks?

**Textual temporal reasoning transfer:** Table 2 presents the evaluation results of models trained on various compositions of our datasets. Continual training with the original LLaVA-Next dataset yields negligible improvements, while HotpotQA training surprisingly degrades performance, despite its similarity to temporal location tasks. These findings underscore the non-trivial nature of enhancing temporal understanding in Video LLMs. Our textual temporal transfer approach, by contrast, demonstrates significant improvements. Regarding the different tasks, we find that: (i) The Temporal Change (1x) subset which combines Order-GPT (1x) and Attribute (1x), enhances Order accuracy from 54.3 to 65.2 and Attribute accuracy from 51.7 to 63.9. Increasing the number of distractor captions (1x to 8x) leads to generally better performance. (ii) Order-Template and Temporal Referring excel in the Speed dimension, while Temporal Grounding negatively influences all aspects. (iii) Combining different aspects often leads to synergistic improvements. For example, supplementing Temporal Change (1x) with other tasks leads to better results than training with the set alone. This suggests complementary benefits across diverse temporal reasoning skills. Finally, despite without any video data, our T3 composition with all synthesized tasks, helps LongVA-7B achieve the best overall accuracy, even outperforming ShareGPT4Video-8B trained on 28,000 video samples annotated by GPT-4V. These results highlight the effectiveness of our approach in improving Video LLMs' temporal reasoning across various dimensions.

**Holistic video understanding evaluation:** Based on previous results, we adopt the T3 composition, which achieves the highest textual validation scores, to explore transfer effects on comprehensive video understanding benchmarks. Table 3 shows detailed task accuracy results on MLVU. Further training on the original image-text instruction dataset or Hotpot QA decreased overall performance, indicating that video temporal understanding cannot be enhanced through continued fine-tuning or long-context training alone. In contrast, mixing our T3 with LLaVA-Next improves performance across most tasks. Notably, the two highly temporal-oriented tasks, Action Count (AC) and Action Order (AO), see a substantial average gain of 8.3 points. Specifically, AC scores increased by 3.9 points ($25.2 \rightarrow 29.1$), while AO scores improved by 12.7 points ($41.7 \rightarrow 54.4$). Table 4 demonstrates

substantial improvement in Video-MME's temporal reasoning subtask (37.3 → 49.7), even outperforming larger models such as InternVL-Chat-V1.5-20B and VILA1.5-40B. Besides, compared to fine-tuning with LLaVA-Next and Hotpot QA, our method yields the best overall performance at 55.0. Further analysis on distribution overlapping between our synthesized samples and downstream tasks justifies the performance gains are not sourced from simply memorizing the distribution of questions and answer choices D.2. We also observe strong correlations between the textual and video temporal reasoning scores across benchmarks in Appendix D.3. These results validate that enhanced textual temporal reasoning in LLM backbones effectively transfers to holistic video understanding.

**Critical LLM components for temporal understanding:** We evaluate the impact of selectively unfreezing LLM components across TempCompass, MLVU, and Video-MME datasets using the T3 /w LLaVA-Next dataset. Figure 5 illustrates that fully unfreezing the language model yields optimal performance.

For TempCompass, the fully unfrozen model achieves 58.2% accuracy, outperforming self-attention-only (57.7%) and MLP-only (55.1%) configurations. Notably, unfreezing self-attention modules consistently surpasses unfreezing MLP layers, with MLVU showing a substantial 4.2 percentage point difference (55.8% v.s. 51.6%) and a 2.3 absolute accuracy gap on Video-MME (52.4% v.s. 50.1%). This suggests that self-attention modules play a more crucial role in transferring temporal understanding capabilities. These findings align with the understanding that self-attention modules act as dynamic information aggregators (Wang et al., 2023b), while MLP layers primarily serve as static knowledge banks (Geva et al., 2021).

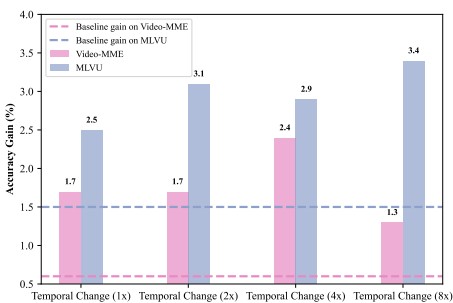

Figure 5: Compared to MLP modules, self-attention modules are more important for temporal reasoning.

**Enhanced utilization of increased video frames:** We examine whether temporal-enhanced LLMs could better leverage increased input video frames by comparing performance gains when increasing frames from 32 to 128, using models trained on Temporal Change sets. Figure 6 shows that our textual temporal reasoning-enhanced models benefit more from increased input frames compared to the baseline LongVA. On TempCompass, the Temporal Change (1x) model gains 1.7% accuracy with 128 frames, versus 0.6% for LongVA. Similarly, on MLVU, enhanced models achieve nearly double the gain of the baseline. These results indicate that our approach enables better utilization of additional input frames. Training on longer inputs with more distractor captions generally yields larger gains, though the optimal length varies between datasets.

**Transfer to Other Temporal Concepts:** Our designed textual temporal tasks focus on four fundamental temporal aspects inspired by previous diagnosis benchmarks (Liu et al., 2024b; Fu et al., 2024). While these aspects may appear basic, they form essential building blocks for complex temporal reasoning and are particularly challenging for current Video LLMs to master, as evidenced in our previous analysis. To rigorously evaluate the generalization capability of our method, particularly to temporal concepts that are challenging to describe textually (e.g., precise rotation counts, relative motion speeds), we test our models on two recent benchmarks with complementary temporal characteristics: TOMATO (Shangguan et al., 2024) and Vinoground (Zhang et al., 2024a). TOMATO evaluates fine-grained motion understanding that goes beyond discrete textual descriptions, testing continuous aspects of motion dynamics. Our T3-enhanced model shows significant improvements in velocity and frequency understanding (Vel. & Freq) (39.9 vs. 23.1, + 16.8) and Shape & Trend comprehension (30.5 vs. 22.4, + 8.1). The decreased performance in Rotation (16.6 vs. 26.4, - 9.8) reveals an inherent limitation: purely

Figure 6: Our temporal-enhanced models leverage more input video frames better.

Table 5: Generalization evaluation on TOMATO and Vinoground benchmarks.

| Model | TOMATO (Shangguan et al., 2024) | | | | | | | Vinoground (Zhang et al., 2024a) | | |
|---|---|---|---|---|---|---|---|---|---|---|
| | Rotation | Direction | Vel. & Freq. | Shape & Trend | Visual Cues | Action Count | All | Text | Video | Group |
| LongVA-7B | 26.4 | 21.6 | 23.1 | 22.4 | 38.6 | 14.7 | 22.3 | 21.2 | 21.6 | 5.4 |
| + LLaVA-Next | 22.2 (-4.2) | 25.6 (+4.0) | 21.2 (-1.9) | 26.0 (+3.6) | 40.0 (+1.4) | 19.8 (+5.1) | 21.2 (-1.1) | 19.0 (-2.2) | 21.8 (+0.2) | 4.6 (-0.8) |
| + T3 w/ LLaVA-Next (Ours) | 16.6 (-9.8) | 21.6 (-) | 39.9 (+16.8) | 30.5 (+8.1) | 38.6 (-) | 16.4 (+1.7) | 24.3 (+2.0) | 31.6 (+10.4) | 30.0 (+8.4) | 11.8 (+6.4) |

text-based training may not fully capture precise rotational movements that require continuous, quantitative understanding. On the other hand, Vinoground examines temporal reasoning ability in real-world scenarios through carefully curated video-caption pairs that share identical static content but differ in temporal action sequences. For instance, distinguishing between *cut then turn* versus *turn then cut* requires a precise understanding of temporal order. Our method substantially improves LongVA-7B's performance across all metrics: Text (31.6, +10.4), Video (30.0, +8.4), and Group (11.8, +6.4). These comprehensive evaluations demonstrate that our T3 develops robust temporal reasoning abilities that generalize well to continuous motion understanding (TOMATO) and different task formats (Vinoground).

## 5 RELATED WORK

**Video Large Language Models (Video LLMs).** Video LLMs, integrating LLMs and visual encoders, have shown promising results on diverse video tasks (Tang et al., 2023a). These models typically leverage open-source LLMs (Touvron et al., 2023; Yang et al., 2024) for generation and reasoning capabilities. Recent architectural explorations have focused on efficient video encoding, including Video Transformer and Q-Former (Li et al., 2023b; Ren et al., 2023), spatial and temporal QFormer (Zhang et al., 2023; Tan et al., 2024), and memory bank for long video frames (He et al., 2024). Other approaches like Video-LLaVA (Lin et al., 2023a), Chat-UniVi (Jin et al., 2023), VILA-series (Lin et al., 2023b), and LLaVA-series (Liu et al., 2024a; Zhang et al., 2024b) demonstrate effective transfer from image to video tasks with image-video unification. In contrast to these studies, our work focuses on probing and enhancing the temporal understanding ability of Video LLMs, identifying the LLM's poor grasp of temporal concepts as a key bottleneck. Our textual-only temporal reasoning transfer effectively addresses this issue without using any image/video instruction tuning data.

**Temporal Understanding Benchmarks for Video LLMs.** Video temporal understanding benchmarks have evolved rapidly to guide Video LLM development. Pilot studies incorporate existing tasks (Wang et al., 2019; Li et al., 2021) into comprehensive assessments (Li et al., 2023c; Ning et al., 2023; Li et al., 2023a). As Video-LLMs become stronger, benchmarks such as EgoSchema (Mangalam et al., 2023), Neptune (Nagrani et al., 2024), Video-MME (Fu et al., 2024) and MLVU (Zhou et al., 2024), focus on stress-testing models with diverse and challenging tasks of long videos. Specific benchmarks addressing temporal understanding (Li et al., 2023d; Liu et al., 2024b; Wang et al., 2023a; Shangguan et al., 2024; Zhang et al., 2024a) have highlighted limitations in current Video LLMs. Our work not only diagnoses the temporal understanding bottleneck in Video LLMs but also demonstrates a correlation between textual and fine-grained video temporal understanding. Importantly, our proposed methods show improvements on challenging benchmarks such as Video-MME, advancing the field of video temporal reasoning.

## 6 CONCLUSIONS AND LIMITATIONS

Our work investigates the temporal reasoning bottleneck of Video LLMs, identifying the language model backbone as the primary source. To address this, we develop a textual temporal reasoning transfer framework that synthesizes QA pairs on various temporal concepts from image-text pairs. Experimental results validate the correlation between textual and visual temporal understanding, demonstrating the efficacy of our method on comprehensive video understanding benchmarks. **Limitations: Limited temporal concept scope:** While our framework addresses four key temporal dimensions and shows potential across temporal tasks, temporal reasoning aspects such as motion-related changes still need further investigation. (ii) **Scalability with model size:** Although T3 improves even large models such as Qwen2-VL-72B, the gains vary across the backbone model scales and tasks. Future work could explore more complex temporal scenarios and develop advanced augmentation pipelines for larger models.

## ACKNOWLEDGMENTS

We gratefully acknowledge the HKU-NLP group and PKU Lanco group for their constructive feedback, and thank the anonymous reviewers for their insightful comments that substantially improved this work. This research was partially supported by the National Natural Science Foundation of China under Grant No. 92470205 and No. 62176002, and the joint research scheme of the National Natural Science Foundation of China (NSFC) and the Research Grants Council (RGC) under grant number N_HKU714/21.

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

# A   DETAILS OF VIDEO LLM TEMPORAL ANALYSIS

## A.1   VIDEOS

§2.2 describes our basic idea to create temporally contradicting videos for different temporal aspects. In this section, we provide more details of the source videos and the composition of synthesized videos. Table 7 summarizes the video composition for different temporal aspects.

### A.1.1   ORDER

**Source Videos.** This aspect encompasses three categories of source videos: people, cats, and flowers. We obtained these videos from the ShutterStock[2] platform. To create distinct training and test sets for the classifier probe, we collected videos of each category with both black and white backgrounds. The black background videos were used for training the probe, while the white background videos were reserved for evaluation.

**Composition of Synthesized Videos.** The synthesized video for this aspect concatenate two or three source videos. As illustrated in Table 7, the "Two Events" videos comprise two categories, showcasing cats and people in reversing sequential orders. The "Three Events" videos expand to six categories, covering all possible permutations of *cat*, *person*, and *flower* sequences.

### A.1.2   ATTRIBUTE

**Source Videos.** This aspect is further divided into *Shape* and *Brightness*. For *Shape*, we collected videos from ShutterStock that illustrate the process of flower blooming. Similar to the *Order* aspect, we gathered flowers with both black and white backgrounds to create separate training and testing sets for the classifier probe. For *Brightness*, we collected static images from the COCO dataset (Lin et al., 2014) and adjusted pixel values to synthesize videos with brightness variations. Car images were used to create training videos, while cat images were used for testing videos.

**Composition of Synthesized Videos.** Videos of *Shape* consist of two categories: flower blooming and its reversed process, with the latter created by inverting the source videos. *Brightness* videos also comprise two categories: brightening and darkening.

### A.1.3   TEMPORAL REFERRING

**Source Videos.** This aspect utilizes the "Three Events" videos previously collected for the *Order* aspect.

**Composition of Synthesized Videos.** The *Temporal Referring* questions are categorized into three types, each referring to a specific part of the video: the beginning, middle, and end. For instance, a typical question might be, "*Which item is shown at the beginning/middle/end of the video?*" For each of these temporal reference types, the videos are further classified into three categories, yielding different possible answers: a person, a cat, or a flower.

### A.1.4   TEMPORAL GROUNDING

**Source Videos.** This aspect also employs the "Three Events" videos originally gathered for the *Order* aspect.

**Composition of Synthesized Videos.** In this aspect, three question types are formulated, i.e., "*In which part of the video can we see a person/cat/flower?*" For each question type, the videos are divided into three categories, resulting in different answers grounding to the video's different temporal locations: beginning, middle, or end.

## A.2   FRAME CAPTIONS

To evaluate the textual temporal understanding capabilities of LLMs, we must ensure that frame captions contain adequate information about relevant temporal aspects. For example, in videos

---

[2]https://www.shutterstock.com

depicting a transition from a cat to a person, the initial frame captions should clearly identify the subject as a cat, while later frame captions should describe the person.

In our approach, we employ a "differential captioning strategy" for aspects such as *Order*, *Shape Attribute*, *Temporal Referring* and *Temporal Grounding*, drawing inspiration from ShareGPT4Video (Chen et al., 2024). As outlined in Table 8,we begin by generating a caption for the first frame. For subsequent frames, captions are created based on both the visual features of the current frame and the captions of preceding frame. Table 10, 11, 12 demonstrate that this method produces captions with sufficient and accurate per-frame information to comprehend these temporal aspects, thanks to the powerful visual perception ability of GPT-4o.

In terms of the *Brightness Attribute*, we find that simple differential captioning cannot accurately reflect brightness changes across video frames due to the lack of a consistent brightness standard. To address this issue, we developed a chain-of-thought (CoT) captioning strategy. This approach first prompts GPT-4o to categorize frame brightness into four levels (very dark, slightly dark, normal, and bright) before generating the caption. As illustrated in Table 11, the resulting frame captions accurately reflect brightness changes throughout the video.

### A.3    CLASSIFIER PROBE TRAINING

**Model Architecture.** Our classifier probe is built upon a single-layer unidirectional LSTM model with a hidden dimension of 128. This probe is trained on visual features that are down-sampled and projected into the LLM embedding space, i.e., the embedding of visual tokens for Video LLMs. Let $\mathbf{V} \in \mathcal{R}^{n \times d_v}$ represent these visual features, where $n = l \times h \times w$ and $l, h, w, d_v$ correspond to the temporal, height, width and channel dimensions, respectively. For an 8-frame input video, the default setup of LongVA-7B is $n = 1152, d_v = 3584$, while for VILA-8B, it is $n = 1568, d_v = 4096$. We further down-sample $\mathbf{V}$ to $\mathbf{V}' \in \mathcal{R}^{n' \times d_v'}$ using bi-linear interpolation, where $n' = 128$ and $d_v' = 1024$. The LSTM probe is then trained on this down-sampled representation $\mathbf{V}'$.

**Data.** Table 7 provides a comprehensive breakdown of the training and test video compositions. For the *Brightness Attribute*, we generated training videos using car images, while the test videos are constructed from cat images. Regarding other temporal aspects, we created training videos with a black background, contrasting with the white background used in the test videos.

**Training Hyper-parameters.** Table 13 shows the training hyper-parameters for the LSTM classifier probe. For relatively easy temporal aspects, i.e., the "Two Events" *Order*, we train the probe for only 15 epoches. For *Attribute* and other aspects, we increase the training epoch to 80 and 120, respectively.

### A.4    RESULTS

### A.4.1    SUB-CATEGORY RESULTS

Figures 7 and 8 provide a comprehensive breakdown of our temporal analysis results. The data reveals significant variations in model performance across different sub-categories within particular temporal aspects. In the *Order* aspect, we observe a stark contrast in difficulty between two-event and three-event sequences. The task of discerning the order of three events proves substantially more challenging, with LongVA-7B and VILA-8B experiencing dramatic accuracy drops of 41.8 and 51.1 points, respectively. Regarding *Temporal Referring*, the accuracy is significantly lower when referring to the middle of the video. The deficiency of the LLM decoder in temporal understanding is also more pronounced in these challenging scenarios.

### A.4.2    RESULTS OF PROBING LLM FEATURES

We also investigate the performance of probing LLM last-layer features, instead of probing the visual features in the embedding space. Specifically, we input videos into LongVA-7B and extract the hidden states from the last layer of the LLM decoder. Two types of probes were then employed: *LSTM Probe*: Trained on all hidden state features, this probe assesses whether temporal information is retained across all the last-layer LLM features. *Linear Probe*: Trained on the hidden state of the last time step, this probe evaluates if the temporal information persists in a single hidden state.

Table 6: Ablation study of the distractor captions and temporal reasoning transfer via images instead of captions.

| Method | Action | Direction | Speed | Order | Attribute Change | Avg. |
|---|---|---|---|---|---|---|
| LongVA-7B (32 frm) | **92.3** | 37.3 | 42.0 | 54.3 | 51.7 | 55.9 |
| + Temporal Change (w/o Distractor) | 90.8 | 40.3 | 41.0 | 61.6 | 60.4 | 59.0 |
| + Temporal Change (1x) | 91.1 | 37.3 | 39.8 | 65.2 | 63.9 | 59.5 |
| + Temporal Change (2x) | 89.6 | 39.1 | 40.4 | **66.9** | 61.8 | 59.6 |
| + Temporal Change (4x) | 90.5 | **43.0** | 39.4 | 64.2 | 64.2 | 60.4 |
| + Temporal Change (8x) | 90.2 | 42.4 | 39.8 | 64.2 | **68.1** | **61.0** |
| + Image Order | 86.4 | 36.1 | 40.4 | 55.3 | 56.9 | 55.2 |
| + Textual Order | 91.4 | 38.2 | 41.6 | 63.6 | 58.7 | 58.9 |

Table 14 presents the probing results. We can see that the LSTM probe's accuracy on LLM features is consistently lower than that on visual features across all temporal aspects, especially for the Attribute aspect (88.9% vs. 73.9%). The accuracy of the linear probe is even lower. These findings demonstrate that a considerable portion of the temporal information is lost when transitioning from visual features to the final-layer LLM features. Furthermore, we observe a substantial gap between the performance of the last-layer LLM features and the final output of LongVA. This suggests that even when relevant information is present in the LLM hidden states, the Video LLM can still produce incorrect answers. Investigating the causes of this discrepancy is left for future work.

# B  DETAILS OF TEXTUAL TEMPORAL UNDERSTANDING DATA

## B.1  CAPTION POOL

The contextual information of our textual QA data are sourced from the detailed image captions from the LLaVA-ReCap-558K dataset [3]. These captions are generated by the LLaVA-Next-34B model (Liu et al., 2024a) based on images from the LCS-558K (a subset of the LAION/CC/SBU dataset). We only retain the first sentence of the detailed image captions to form the caption pool $\mathbf{C}_{\mathrm{pool}}$.

## B.2  TEXTUAL TEMPORAL REASONING DATA CONSTRUCTION

The construction process of our textual temporal reasoning datasets are illustrated in detail in Algorithm 1, 2, 3, 4, 5. The prompts used to generate question-answer pairs and captions are shown in Table 16, 17, 18, 19, 20.

## B.3  DATASET STATISTICS

Table 15 presents detailed statistics of our textual temporal datasets and the two baseline datasets. The information includes the number of samples, the number of relevant and distractor caption, the number of input/output tokens, and the involved modalities.

## B.4  ABLATION STUDY OF TEXTUAL TEMPORAL REASONING TRANSFER

**Effect of distractor captions.**    We maintain the same setting as in our main paper and compare the results of the Temporal Change sets with and without distractor captions. As shown in the middle block of Table 6, incorporating distractor captions yields a more pronounced improvement in the targeted Order and Attribute Change aspects. Moreover, increasing the number of distractor captions generally leads to better performance. These results validate our approach of inserting confusing captions to enhance textual temporal reasoning transfer.

**Transfer via images versus captions.**    To investigate this difference, we replace the caption with the original image in our Order set, keeping all other settings constant. As shown in the bottom block of

---

[3]https://huggingface.co/datasets/lmms-lab/LLaVA-ReCap-558K

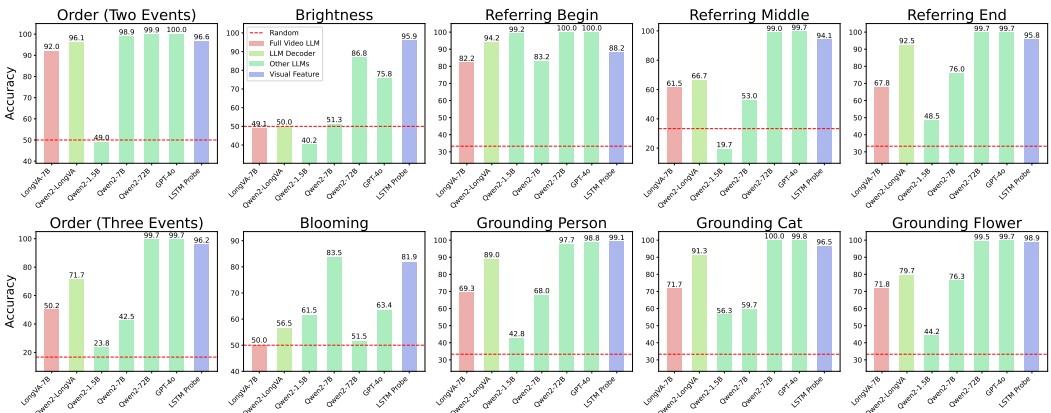

Figure 7: Detailed temporal analysis results of LongVA.

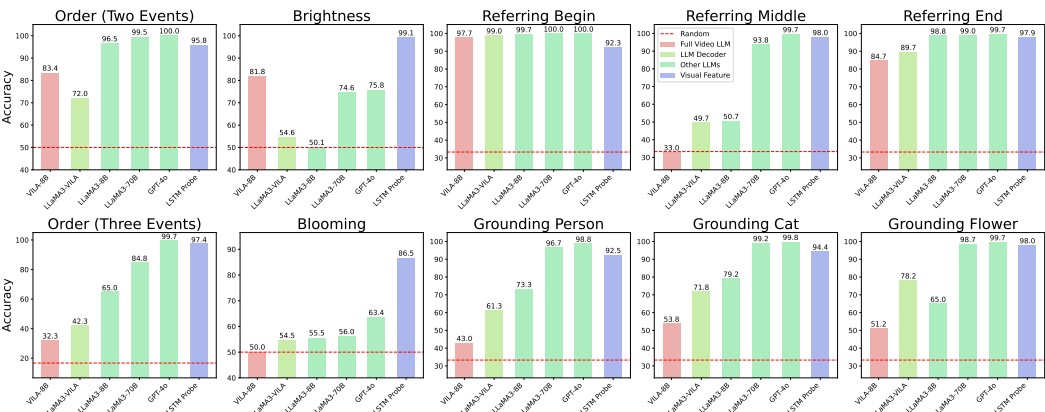

Figure 8: Detailed temporal analysis results of VILA.

Table 6, textual temporal reasoning outperforms the corresponding transfer set using images (Image Order). Notably, transfer via images only marginally increases the Order performance from 54.3 to 55.4, while text format transfer significantly boosts it to 63.6. This substantial gap corroborates our findings in the main paper that the temporal reasoning bottleneck lies on the LLM's side, and therefore, the textual format is more effective in enhancing temporal reasoning capability.

---

**Algorithm 1:** Textual temporal QA generation for the *Order* aspect using GPT-4-turbo. $\text{ExtCont}(\mathbf{C}_{\text{pool}}, \mathbf{C}, \text{n})$ inserts relevant captions $\mathbf{C}$ in between $n$ distractor captions and ensures that every pair of relevant and distractor caption do not share common nouns.

**Input:** Caption pool $\mathbf{C}_{\text{pool}}$, number of relevant captions $n_{\text{rcap}}$, number of distractor captions $n_{\text{dcap}}$, large language model $\text{LLM}()$, function to extend context $\text{ExtCont}(\mathbf{C}_{\text{pool}}, \mathbf{C}, \text{n})$, QA generation prompt $\mathbf{p}$

**Output:** SFT data sample $\{\mathbf{x}_{\text{in}}, \mathbf{x}_{\text{out}}\}$

1 # Sample relevant captions
2 $\mathbf{C}_{\text{r}} = \{C_{\text{r}}^1, C_{\text{r}}^2, ..., C_{\text{r}}^{n_{\text{rcap}}}\} \sim \mathbf{C}_{\text{pool}}$
3 # Create extended context
4 $\mathbf{C}^{ext} = \text{ExtCont}(\mathbf{C}_{\text{pool}}, \mathbf{C}_{\text{r}}, n_{\text{dcap}})$
5 # Generate questions and answers
6 $Q, A = \text{LLM}(\mathbf{C}_{\text{r}}, \mathbf{p})$
7 # Concatenate context and question
8 $\mathbf{x}_{\text{in}} = \mathbf{C}^{ext} \oplus Q, \mathbf{x}_{\text{out}} = A$

Table 7: Details of the videos used for our temporal analytical study in Section 2. "A→B" denotes a synthesized video first showing A and then B. (black) and (white) indicate videos with black and white background, respectively. (car) and (cat) indicate videos showing cars and cats, respectively. xN denotes the number of videos. "Train Videos" is only used when training the classifier probe.

| Dataset | # Classes | Train Videos | Test Videos |
|---|---|---|---|
| **Order** | | | |
| Two Events | 2 | c1: person→cat (black) x400
c2: cat→person (black) x400 | c1: person→cat (white) x400
c2: cat→person (white) x400 |
| Three Events | 6 | c1: person→cat→flower (black) x400
c2: person→flower→cat (black) x400
c3: cat→person→flower (black) x400
c4: cat→flower→person (black) x400
c5: flower→person→cat (black) x400
c6: flower→cat→person (black) x400 | c1: person→cat→flower (white) x100
c2: person→flower→cat (white) x100
c3: cat→person→flower (white) x100
c4: cat→flower→person (white) x100
c5: flower→person→cat (white) x100
c6: flower→cat→person (white) x100 |
| **Attribute** | | | |
| Shape | 2 | c1: blooming (black) x177
c2: unblooming (black) x177 | c1: blooming (white) x100
c2: unblooming (white) x100 |
| Brightness | 2 | c1: brightening (car) x955
c2: darkening (car) x955 | c1: brightening (cat) x394
c2: darkening (cat) x394 |
| **Temporal Referring** | | | |
| Begin | 3 | c1: {person→cat→flower (black) x400,
person→flower→cat (black) x400}
c2: {cat→person→flower (black) x400,
cat→flower→person (black) x400}
c3: {flower→person→cat (black) x400,
flower→cat→person (black) x400} | c1: {person→cat→flower (white) x100,
person→flower→cat (white) x100}
c2: {cat→person→flower (white) x100,
cat→flower→person (white) x100}
c3: {flower→person→cat (white) x100,
flower→cat→person (white) x100} |
| Middle | 3 | c1: {cat→person→flower (black) x400,
flower→person→cat (black) x400}
c2: {person→cat→flower (black) x400,
flower→cat→person (black) x400}
c3: {person→flower→cat (black) x400,
cat→flower→person (black) x400} | c1: {cat→person→flower (white) x100,
flower→person→cat (white) x100}
c2: {person→cat→flower (white) x100,
flower→cat→person (white) x100}
c3: {person→flower→cat (white) x100,
cat→flower→person (white) x100} |
| End | 3 | c1: {cat→flower→person (black) x400,
flower→cat→person (black) x400}
c2: {person→flower→cat (black) x400,
flower→person→cat (black) x400}
c3: {person→cat→flower (black) x400,
cat→person→flower (black) x400} | c1: {cat→flower→person (white) x100,
flower→cat→person (white) x100}
c2: {person→flower→cat (white) x100,
flower→person→cat x100}
c3: {person→cat→flower (white) x100,
cat→person→flower (white) x100} |
| **Temporal Grounding** | | | |
| Person | 3 | c1: {person→cat→flower (black) x400,
person→flower→cat (black) x400}
c2: {cat→person→flower (black) x400,
flower→person→cat (black) x400}
c3: {cat→flower→person (black) x400,
flower→cat→person (black) x400} | c1: {person→cat→flower (white) x100,
person→flower→cat (white) x100}
c2: {cat→person→flower (white) x100,
flower→person→cat (white) x100}
c3: {cat→flower→person (white) x100,
flower→cat→person (white) x100} |
| Cat | 3 | c1: {cat→person→flower (black) x400,
cat→flower→person (black) x400}
c2: {person→cat→flower (black) x400,
flower→cat→person (black) x400}
c3: {person→flower→cat (black) x400,
flower→person→cat (black) x400} | c1: {cat→person→flower (white) x100,
cat→flower→person (white) x100}
c2: {person→cat→flower (white) x100,
flower→cat→person (white) x100}
c3: {person→flower→cat (white) x100,
flower→person→cat (white) x100} |
| Flower | 3 | c1: {flower→person→cat (black) x400,
flower→cat→person (black) x400}
c2: {person→flower→cat (black) x400,
cat→flower→person (black) x400}
c3: {person→cat→flower (black) x400,
cat→person→flower (black) x400} | c1: {flower→person→cat (white) x100,
flower→cat→person (white) x100}
c2: {person→flower→cat (white) x100,
cat→flower→person (white) x100}
c3: {person→cat→flower (white) x100,
cat→person→flower (white) x100} |

Table 8: The prompt used to generate frame captions for videos of *Order*, *Shape Attribute*, *Temporal Referring* and *Temporal Grounding*.

---

**First Frame:**
You are an advanced AI visual assistant. You will be provided with the first frame extracted from a video clip. Your task is to describe this frame in as much detail as possible, focusing on the following elements:

1. **Key Objects**: Identify and mention the main objects or subjects present in the frame. Be specific and provide relevant details about each object.
2. **Visual Attributes**: Describe the visual characteristics of the key objects, such as their color, size, shape, texture, or any other notable features. Pay special attention to the overall brightness or lighting conditions in the frame.
3. **Location**: Specify the location or setting of the frame, including the background, environment, or any identifiable landmarks or surroundings.
4. **Potential Action**: If applicable, describe any actions or activities that the key objects might be engaged in or are likely to perform based on their positioning, pose, or context within the frame.
5. **Movement**: If there is any visible or implied movement in the frame, describe the direction, trajectory, or nature of the movement for the relevant objects.

Ensure your description accurately reflects only the contents of this frame. Do not reference any part of this prompt in your response. Begin with "This frame". The caption should be written in present tense and should not exceed 2 sentences.

**Other Frames:**
You are an advanced AI visual assistant tasked with describing frames extracted from a video clip. When provided with a frame, describe it in detailed and accurate terms, focusing on the changes of following elements:

1. **Key Objects**: Identify and mention the main objects or subjects present in the frame. Be specific and provide relevant details about each object.
2. **Visual Attributes**: Describe the visual characteristics of the key objects, such as their color, size, shape, texture, or any other notable features. Pay special attention to the overall brightness or lighting conditions in the frame.
3. **Location**: Specify the location or setting of the frame, including the background, environment, or any identifiable landmarks or surroundings.
4. **Potential Action**: If applicable, describe any actions or activities that the key objects might be engaged in or are likely to perform based on their positioning, pose, or context within the frame.
5. **Movement**: If there is any visible or implied movement in the frame, describe the direction, trajectory, or nature of the movement for the relevant objects.

While your primary focus should be the current frame, you can reference the provided caption of the preceding frame to describe any relevant relationships between the two frames. Ensure your description reflects only the contents of the current frame and do not include any elements of this prompt in your response. Begin with "This frame". The caption should be written in present tense and should not exceed 2 sentences.
The current frame is uploaded as an image. The caption of the preceding frame is provided below:
[previous_frame_caption]

---

## C  TRAINING DETAILS

### C.1  TRAINING DATASETS

Table 21 provides details of the training datasets used in our main paper. To ensure fair comparison across all compositions, we set the number of training samples to 22k, which is the maximum

Table 9: The prompt used to generate frame captions for the videos of *Brightness Attribute*. We prompt GPT-4o to determine the level of brightness before generating the frame caption, which ensures information of brightness change is reflected in the frame captions (Table 11 shows an example).

Analyze the provided image and categorize its overall brightness level into one of the following categories: 'bright', 'normal', 'slightly dark', 'very dark'. Based on this assessment, craft a detailed description of the image, reflecting the main elements presented in the image.

Here are two examples of classifed brightness and corresponding descriptions:
{"brightness": "bright",
"description": "A person is sitting on a bench with a book in a bright environment." }
{"brightness": "slightly dark",
"description": "The image shows an slightly dark room with a red balloon." }

Now please give your classification result and description, following the above JSON format:

Table 10: Example of a video focusing on the *Order* aspect and corresponding frame captions used to evaluate the LLM decoder. We show four frames here but in practice eight frames are used. The words in blue highlight the major subject depicted in each frame.

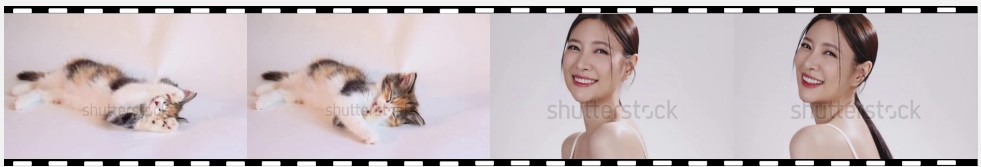

Frame Captions:
Frame 1: This frame shows a fluffy white and gray kitten lying on its side against a plain white background, playing with a small purple toy. The kitten is holding the toy with its front paws and appears to be in a playful mood, with some motion blur suggesting movement.

Frame 3:This frame depicts the same fluffy white and gray kitten, now lying on its side on the plain white background. The kitten's eyes are still closed, and its head is resting on the ground, continuing to appear relaxed or asleep.

Frame 5: This frame shows a woman with smooth blue skin and dark hair tied back, gazing directly at the camera. She is positioned against a plain gray background with visible light reflecting from her skin, and she appears calm and serene based on her expression and posture.

Frame 7: This frame shows a smiling woman with smooth blue skin and dark, straight hair tied back, facing slightly away from the camera with her head turned toward the viewer. She is set against a plain gray background, and her joyful expression and body posture convey a sense of happiness and contentment.

available in the smallest subsets (Referring and Grounding). For the transfer evaluation on MLVU and Video-MME, we combine the original LLaVA-Next image-text SFT dataset with our textually synthesized samples, as shown in Table 22. The mixing ratio and total sample size are determined based on the exploration results discussed in the following subsection.

Table 11: Example of a video focusing on the *Brightness Attribute* aspect and corresponding frame captions used to evaluate the LLM decoder. The key information of brightness change is highlighted in blue.

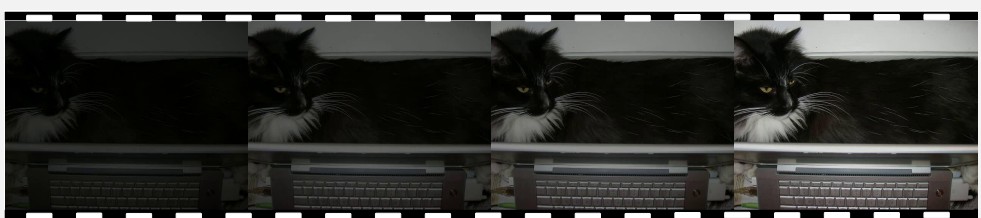

Frame Captions:
Frame 1: The image shows a very dark scene with a black and white cat lying on top of a laptop keyboard. The cat's face and whiskers are slightly visible, but the overall image is hard to see due to the low brightness.

Frame 3: The image shows a black and white cat lying on top of a laptop keyboard. The overall scene is very dark, making it difficult to see details clearly.

Frame 5: The image shows a black and white cat lying on top of a laptop keyboard in a slightly dark environment.

Frame 7: The image shows a black and white cat lying on top of a laptop keyboard. The cat appears to be comfortably settled in the space between the laptop screen and keyboard. The overall brightness of the image is normal, allowing clear visibility of the cat's fur and the laptop's keys.

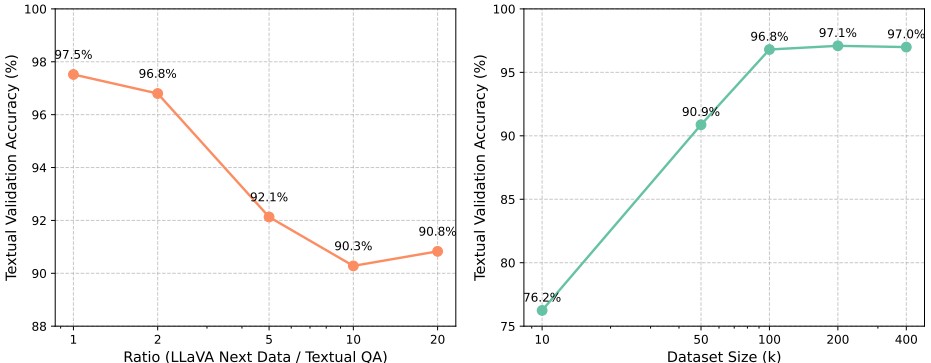

Figure 9: (Left) Exploration of mixing ratios between image-text instruction tuning and textual temporal QA. (Right) Dataset scaling analysis.

## C.2 DATASET MIXING EXPLORATIONS

Our preliminary study revealed that training the model exclusively on large-scale textual temporal QA samples led to catastrophic forgetting of visual perception abilities, degrading it to a text-only model. To mitigate this issue, we explore the integration of original instruction tuning datasets to enhance video understanding transfer performance. Specifically, we combine the original LLaVA-Next dataset with textual temporal QA pairs from our All Temporal subset, varying the mixing ratio from 1 to 20, while fixing total samples to 100k for fair comparison. We use textual validation accuracy as our selection criterion, as it correlates well with temporal understanding ability, as discussed in our main paper.

Table 12: Example of a video focusing on the *Shape Attribute* aspect and corresponding frame captions used to evaluate the LLM decoder. The key information of shape change is highlighted in blue.

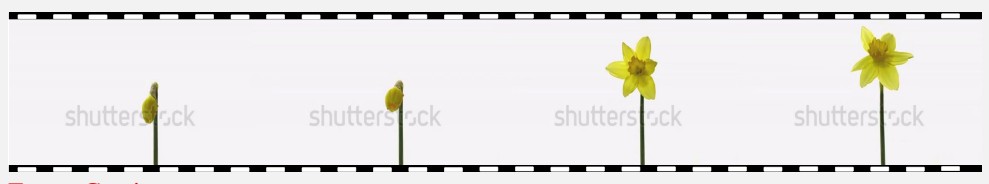

Frame Captions:

Frame 1: This frame shows a blue bud at the end of a green cotton swab positioned vertically against a plain white background. The cotton bud appears clean and unused, with no visible movement or action taking place.

Frame 3: This frame shows the same blue bud at the end of a green cotton swab, still positioned vertically against a plain white background. The cotton on the bud appears slightly more compressed and misshapen compared to the preceding frame.

Frame 5: This frame shows a blue flower with five petals on a green stem, set against a white background. The flower appears to be blooming mid-stage with slightly spread petals, and the stem is straight and upright, positioned centrally in the frame.

Frame 7: This frame features the same blue flower with six petals on a green stem, now slightly to the right against a white background. The petals are still widely spread, and the stem remains upright, suggesting no noticeable movement or change from the previous frame.

Table 13: Training hyper-parameters for the classifier probe.

| Temporal Aspect | Learning Rate | Batch Size | #Epoch | Optimizer |
|---|---|---|---|---|
| Order (Two Events) | 5e-5 | 64 | 15 | Adam |
| Order (Three Events) | 5e-5 | 64 | 120 | Adam |
| Attribute | 5e-5 | 64 | 80 | Adam |
| Temporal Referring | 5e-5 | 64 | 120 | Adam |
| Temporal Grounding | 5e-5 | 64 | 120 | Adam |

Table 14: Results of probing LLM features versus visual features.

| Model | Attribute | Referring | Grounding | Order | Average |
|---|---|---|---|---|---|
| LSTM Probe Visual Features | **88.9** | **92.7** | **98.2** | **96.4** | **94.0** |
| LSTM Probe LLM Features | 73.9 | 87.2 | 91.9 | 94.2 | 86.8 |
| Linear Probe LLM Features | 72.7 | 80.0 | 80.4 | 80.4 | 78.4 |
| LongVA-7B | 49.6 | 70.5 | 70.9 | 71.1 | 65.5 |

Figure 9 (left) illustrates that performance on the textual component gradually decreases as more image-text instruction tuning samples are included. Based on these results, we selected a ratio of 2, as it maintains relatively high textual validation accuracy ($> 95\%$) while incorporating sufficient visual-text samples to preserve the model's visual capabilities.

We further explore the optimal sample size for the mixed dataset, with results shown in Figure 9 (right). As expected, textual accuracy continuously improved with larger training samples. This trend holds for downstream tasks such as MLVU and Video-MME as well as shown in Table 23. Consequently, we set the total sample size to 200k, as this threshold first exhibits saturated textual validation accuracy while keeping the sample size manageable for efficient training.

Table 15: Dataset statistics. 1x∼8x indicates the length of the textual context, controlled by the number of distractor captions. #tokens are counted using LongVA-7B tokenizer.

| Dataset | #Samples | #Relevant Captions | #Distractor Captions | #Input Tokens | #Output Tokens | Modalities |
|---|---|---|---|---|---|---|
| Order-GPT (1x) | 16k | 2∼4 | 100±50 | 1.8k | 13.6 | Text |
| Order-GPT (2x) | 15k | 2∼4 | 200±50 | 3.5k | 13.6 | Text |
| Order-GPT (4x) | 15k | 2∼4 | 400±50 | 6.9k | 13.5 | Text |
| Order-GPT (8x) | 15k | 2∼4 | 800±50 | 13.7k | 13.6 | Text |
| Attribute (1x) | 34k | 2 | 100±50 | 1.8k | 8.4 | Text |
| Attribute (2x) | 15k | 2 | 200±50 | 3.5k | 8.4 | Text |
| Attribute (4x) | 15k | 2 | 400±50 | 6.9k | 8.3 | Text |
| Attribute (8x) | 15k | 2 | 800±50 | 13.7k | 8.3 | Text |
| Order-Template (phrase) | 30k | 3∼6 | 200±50 | 3.6k | 17.1 | Text |
| Order-Template (prefix) | 30k | 3∼6 | 200±50 | 3.6k | 15.8 | Text |
| Order-Template (sentence) | 30k | 3∼6 | 200±50 | 3.6k | 77.1 | Text |
| Referring | 22k | 3 | 200±50 | 3.5k | 4.5 | Text |
| Grounding | 22k | 3 | 200±50 | 3.5k | 8.4 | Text |
| Hotpot QA | 90k | - | - | 1.4k | 4.0 | Text |
| LLaVA-Next | 1M | - | - | 36.5 | 57.7 | Text+Image |

# D ADDITIONAL EXPLORATIONS

## D.1 RESULTS WITH QWEN2-VL-72B

To validate our method's effectiveness on state-of-the-art video language models, we evaluate T3 on Qwen2-VL-72B (Wang et al., 2024), a state-of-the-art Video LLMs trained on large-scale video-text data. Following the best practices provided for fine-tuning, we fine-tune the model using the ms-swift framework (Zhao et al., 2024) with the default LoRA setup. The model is trained with a batch size of 64 for 1000 steps. To preserve the general video understanding capabilities of the model, we combine our T3 dataset with LLaVA-Video-SFT (Zhang et al., 2024d) in a 1:2 ratio, consistent with our previous recipe. As shown in Tables 24 and 25, T3 yields consistent improvements across both 16-frame and 32-frame settings. The gains are particularly pronounced for temporal reasoning tasks on MLVU, e.g., Action Count: +9.2% and Action Order: +8.1%. These results demonstrate that T3's benefits extend to large-scale models, providing complementary temporal reasoning capabilities even when applied to models already pre-trained on video data. The consistent improvements across model scales highlight the generality and effectiveness of our approach.

## D.2 ANALYSIS OF PERFORMANCE GAINS

To validate that our T3 method's improvements stem from enhanced temporal reasoning rather than domain-specific training, we conduct two critical analyses: (i) examination of potential dataset distribution overlap and (ii) investigation of model performance without visual inputs.

First, we analyze the data distribution between our T3 dataset and the video benchmarks, in terms of the entire questions and choices along, respectively. Specifically, our analysis consists of the following steps: (1) We utilize our validation set (each subset contains 500 examples) and randomly sample 500 examples from the three video benchmarks. (2) Similarity is measured using Jaccard Similarity: $J(T1, T2) = \frac{|BoW(T1) \cap BoW(T2)|}{|Bow(T1) \cup BoW(T2)|}$, where "BoW" refers to the bag-of-words representation of a piece of text, excluding stop words. (3) For each validation subset (e.g., *Order*, *Referring*, etc), we calculate the percentage of examples with a Jaccard Similarity greater than 0.5 compared to any example in the video benchmark. The results are shown in Table 26, from which we can see that only a small fraction of T3 dataset examples have similar counterparts in the video benchmarks. This indicates that the performance improvement of our method is due to enhanced temporal reasoning capabilities, rather than simply memorizing the distribution of questions and answer choices.

Second, we evaluate model performance in a text-only setting by removing video inputs. As shown in Tables 27, 29, and 28, LongVA-7B's performance drops significantly without visual context (e.g.,

---

**Algorithm 2:** Textual temporal QA generation for the *Order* aspect using templates.

---

**Input:** Caption pool $\mathbf{C}_{\text{pool}}$, number of relevant captions $n_{\text{rcap}}$, number of distractor captions $n_{\text{dcap}}$, large language model $\text{LLM}()$, target to shuffle $tg \in \{sentence, phrase, prefix\}$, function to extend context $\text{ExtCont}(\mathbf{C}_{\text{pool}}, \mathbf{C}, \text{n})$, function to extract a noun phrase from a sentence $\text{ExtrcPhrase}(C)$, order-related question templates $\mathbf{Q}^{tp}$, prefix templates $\mathbf{P}_{prefix}$

**Output:** SFT data sample $\{\mathbf{x}_{\text{in}}, \mathbf{x}_{\text{out}}\}$

1   `# Sample relevant captions`
2   $\mathbf{C}_{\text{r}} = \{C_{\text{r}}^1, C_{\text{r}}^2, ..., C_{\text{r}}^{n_{\text{rcap}}}\} \sim \mathbf{C}_{\text{pool}}$
3   `# Create extended context`
4   $\mathbf{C}^{ext} = \text{ExtCont}(\mathbf{C}_{\text{pool}}, \mathbf{C}_{\text{r}}, n_{\text{dcap}})$
5   `# Sample a question from the templates, e.g., "Reorder the`
     `following captions according to the above video."`
6   $Q \sim \mathbf{Q}^{tp}$
7   **if** $tg == sentence$ **then**
8     $Q = Q \oplus \text{Shuffle}(\mathbf{C}_r)$ `# Add shuffled captions after the question`
9     $A = \mathbf{C}_r$
10   **end**
11   **if** $tg == phrase \text{ or } tg == prefix$ **then**
12     `# Extract noun phrases from the captions`
13     $\mathbf{P} = \{\text{ExtrcPhrase}(C_r) | C_r \in \mathbf{C}_r\}$
14     `# Shuffle the phrases`
15     $\mathbf{P}_{shuf} = \text{Shuffle}(\mathbf{P})$
16     **if** $tg == prefix$ **then**
17       `# Add prefix before the phrases, e.g., (1)(2)(3)`
18       $\mathbf{P}_{shuf} = \{p_{pfx} \oplus p_{pha} | p_{pha} \in \mathbf{P}_{shuf}, p_{pfx} \in \mathbf{P}_{prefix}\}$
19       $\mathbf{P} = \{p_{pfx} \oplus p_{pha} | p_{pha} \in \mathbf{P}, p_{pfx} \in \mathbf{P}_{prefix}\}$
20     **end**
21     `# Add shuffled phrases after the question`
22     $Q = Q \oplus \mathbf{P}_{shuf}$
23     `# Sample three other permutations for form the options`
24     $\mathbf{o} = \mathbf{P} \oplus \text{Sample}(\text{Permutations}(\mathbf{P}) \setminus \mathbf{P}, 3)$
25     $Q = Q \oplus \text{Shuffle}(\mathbf{o})$
26     $A = \mathbf{P}$
27   **end**
28   `# Concatenate context and question`
29   $\mathbf{x}_{\text{in}} = \mathbf{C}^{ext} \oplus Q, \mathbf{x}_{\text{out}} = A$

---

from 55.9% to 41.5% on TempCompass). Notably, our T3 training provides no improvement in this text-only scenario (41.3%). This finding strongly suggests that our method's effectiveness comes from genuine enhancement of temporal reasoning capabilities rather than memorization of answer patterns or question distributions.

## D.3   Correlation between Textual and Video Temporal Understanding

To assess transferability more intuitively, we calculate correlations between textual validation accuracy and benchmark scores of models trained on our textual samples. Figure 10 illustrates a strong correlation with TempCompass overall accuracy (Pearson $r = 0.89$, $p < 0.01$), and MLVU accuracy (Pearson $r = 0.85$, $p < 0.01$). It also positively correlates with Video-MME performances (Pearson $r = 0.59$, $p < 0.05$). These findings validate the feasibility of enhancing video temporal understanding from the LLM side.

---

**Algorithm 3:** Textual temporal QA generation for the *Attribute* aspect.

---

**Input:** Caption pool $\mathbf{C}_{\text{pool}}$, number of distractor captions $n_{\text{dcap}}$, large language model LLM(), set of attributes $\mathbf{a} = \{color, light, shape, posture, emotion\}$, attribute modification prompt $\mathbf{p}_a$, QA generation prompt $\mathbf{p}_{qa}$, function to extend context $\text{ExtCont}(\mathbf{C}_{\text{pool}}, \mathbf{C}, \text{n})$

**Output:** SFT data samples $\{\mathbf{x}_{\text{in}}, \mathbf{x}_{\text{out}}\}$

1 # Sample a caption from the caption pool
2 $C_r \sim \mathbf{C}_{\text{pool}}$
3 # Generate similar captions by modifying color, light condition, shape, posture, or emotion
4 $\mathbf{C}_{attr} = \{C_{attr}|attr \in \mathbf{a}\} = \text{LLM}(C_r, \mathbf{p}_a)$
5 # Create extended context
6 $\mathbf{C}^{ext} = \{\mathbf{C}^{ext}_{attr}|attr \in \mathbf{a}\}$, where $\mathbf{C}^{ext}_{attr} = \text{ExtCont}(\mathbf{C}_{\text{pool}}, C_r \oplus C_{attr}, n_{\text{dcap}})$
7 # Generate questions and answers
8 $\mathbf{Q}, \mathbf{A} = \text{LLM}(\mathbf{C}_{\text{attr}}, \mathbf{p}_{qa})$, where $\mathbf{Q} = \{Q_{attr}|attr \in \mathbf{a}\}$, $\mathbf{A} = \{A_{attr}|attr \in \mathbf{a}\}$
9 # Concatenate context and question
10 $\mathbf{x}_{\text{in}} = \{\mathbf{C}^{ext}_{attr} \oplus Q_{attr}|attr \in \mathbf{a}\}$, $\mathbf{x}_{\text{out}} = \mathbf{A}$

---

**Algorithm 4:** Textual temporal QA generation for the *Temporal Referring* aspect.

---

**Input:** Caption pool $\mathbf{C}_{\text{pool}}$, number of distractor captions $n_{\text{dcap}}$, large language model LLM(), function to extend context $\text{ExtCont}(\mathbf{C}_{\text{pool}}, \mathbf{C}, \text{n})$, element types $\mathbf{e} = \{object, action, attribute\}$, temporal location reference $\mathbf{t} = \{$at the begin, at the middle, at the end$\}$, caption generation prompt $\mathbf{p}_c$, QA generation prompt $\mathbf{p}_{qa}$

**Output:** SFT data sample $\{\mathbf{x}_{\text{in}}, \mathbf{x}_{\text{out}}\}$

1 # Sample a caption from the caption pool
2 $C_r \sim \mathbf{C}_{\text{pool}}$
3 # Generate three similar captions for each element
4 $\mathbf{C} = \{\mathbf{C}_e|e \in \mathbf{e}\} = \text{LLM}(C_r, \mathbf{p}_c)$, where $|\mathbf{C}_e| = 3$
5 # Create extended context
6 $\mathbf{C}^{ext} = \{\mathbf{C}^{ext}_e|e \in \mathbf{e}\}$, where $\mathbf{C}^{ext}_e = \text{ExtCont}(\mathbf{C}_{\text{pool}}, \mathbf{C}_e, n_{\text{dcap}})$
7 # Generate questions and answers. For each element, the three captions share the same question but have different answers.
8 $\mathbf{Q}, \mathbf{A} = \text{LLM}(\mathbf{C}, \mathbf{p}_{qa})$, where $\mathbf{Q} = \{Q_e|e \in \mathbf{e}\}$, $\mathbf{A} = \{\{A|A \in \mathbf{A}_e\}|e \in \mathbf{e}\}$, $|\mathbf{A}_e| = 3$
9 # Add temporal reference to the questions and concatenate the context
10 $\mathbf{x}_{\text{in}} = \{\{\mathbf{C}^{ext}_e \oplus Q_e \oplus t|t \in \mathbf{t}\}|e \in \mathbf{e}\}$, $\mathbf{x}_{\text{out}} = \mathbf{A}$

---

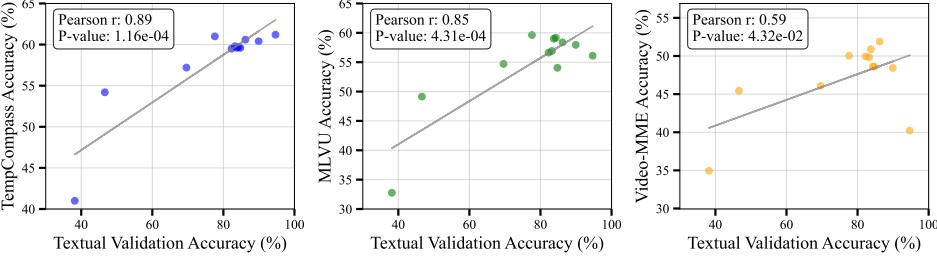

Figure 10: Textual temporal reasoning accuracy correlates positively with video understanding results on three benchmarks.

**Algorithm 5:** Textual temporal QA generation for the *Temporal Grounding* aspect.

**Input:** Element types $\mathbf{e} = \{object, action, attribute\}$, similar captions $\mathbf{C} = \{\mathbf{C}_e | e \in \mathbf{e}\}$, extended context $\mathbf{C}^{ext} = \{\mathbf{C}_e^{ext} | e \in \mathbf{e}\}$, questions for the captions $\mathbf{Q} = \{Q_e | e \in \mathbf{e}\}$, answers to the questions $\mathbf{A} = \{\{A | A \in \mathbf{A}_e\} | e \in \mathbf{e}\}$, large language model LLM(), declarative statement generation prompt $\mathbf{p}_s$, grounding question templates $\mathbf{Q}^{tp} = \{Q_i^{tp}\}_{i=1}^{|\mathbf{Q}^{tp}|}$

**Output:** SFT data sample $\{\mathbf{x}_{\text{in}}, \mathbf{x}_{\text{out}}\}$

1   `# Initialize statement set`
2   $\mathbf{S} = \{\}$
3   **for** $e \in \mathbf{e}$ **do**
4     `# Generate three declarative statements, corresponding to three answers in` $\mathbf{A}_e$
5     $\mathbf{S}_e = \text{LLM}(Q_e, \mathbf{A}_e, \mathbf{p}_s)$, where $|\mathbf{S}_e| = 3$
6     $\mathbf{S} = \mathbf{S} \oplus \mathbf{S}_e$
7   **end**
8   `# Initialize SFT data sample set`
9   $\mathbf{x}_{\text{in}}, \mathbf{x}_{\text{out}} = \{\}, \{\}$
10   **for** $e \in \mathbf{e}$ **do**
11     **for** $S \in \mathbf{S}_e$ **do**
12       `# Sample a grounding question template, e.g., "In which part of the video can we observe [X]?" and insert the statement in it.`
13       $\mathbf{x}_{\text{in}} = \mathbf{x}_{\text{in}} \cup \{\mathbf{C}_e^{ext} \oplus \text{Replace}(Q, [\text{X}], S)\}$, where $Q \sim \mathbf{Q}^{tp}$
14       $\mathbf{x}_{\text{out}} = \mathbf{x}_{\text{out}} \cup \{A\}$, where $A \in \{\text{at the begin, at the middle, at the end}\}$ is determined by the location of corresponding caption in $\mathbf{C}_e^{ext}$
15     **end**
16   **end**

Table 16: The prompt used to generate textual temporal QA for the *Order* aspect.

You will be presented with a list of captions describing keyframes of a video. Your task is to generate five multi-choice questions (and corresponding answers) based on the captions. The questions should focus on the sequential order of the keyframe contents. Make sure that the questions are related to the order of the keyframes and diverse.

Here is an example of captions and the corresponding questions and answers:
Captions:
1. The image shows a person playing basketball.
2. The image shows a dog running on the grass.
3. The image is about a beautiful flower on the table.
4. The image illustrates a bustling city street.

Questions and Answers:
{ "qas": [
    { "question": "Sort the events from the video by their chronological order. (1) city street; (2) dog running on the grass; (3) person playing basketball; (4) flower on the table.",
        "options": [ "(1)(2)(3)(4)", "(3)(2)(4)(1)", "(2)(1)(4)(3)", "(1)(4)(3)(2)" ],
        "answer": "(3)(2)(4)(1)" },
    { "question": "Organize the listed events from the video according to their time sequence: (1) city street (2) dog running on the grass (3) person playing basketball (4) flower on the table",
        "options": [ "city street → dog running on the grass → person playing basketball → flower on the table", "person playing basketball → dog running on the grass → flower on the table → city street", "dog running on the grass → city street → flower on the table → person playing basketball", "city street → flower on the table → person playing basketball → dog running on the grass" ],
        "answer": "person playing basketball → dog running on the grass → flower on the table → city street" },
    { "question": "What is the correct order that objects appear in the video?",
        "options": [ "dog, flower, person, street", "person, dog, flower, street", "street, flower, person, dog", "flower, street, dog, person" ],
        "answer": "person, dog, flower, street" },
    { "question": "In what sequence do the events occur in the video?",
        "options": [ "a dog running and then a person playing basketball", "a city street is shown and then a flower is shown", "a flower appears followed by a city street", "a city street appears followed by a dog runnin ],
        "answer": "a flower appears followed by a city street" }
    ]
}

Now please generate five question-answer pairs based on the following captions. Ensure your resonse follows the above JSON format and style of the example question-answer pairs.
Captions:
**[image_captions]**
Questions and Answers:

Table 17: The prompt used to generate similar captions by modifying attributes in the original caption.

You will be presented with an original image caption. Your task is to enrich this caption with details regarding the attributes of objects or environment. The attributes could include but not limited to these aspects: 'light condition', 'color', 'size & shape', 'emotion' and 'posture'. You are required to create two distinct captions for each attribute. These two captions should contrast each other in terms of the corresponding attribute (e.g., black versus white for 'color', small versus big for 'size & shape').

Here are some examples of original caption and enriched captions related to different aspects:

Original Caption: The image shows a person sitting on the chair.
Enriched Captions:
{ "captions": { "light condition": [ "The image shows a person sitting on the chair with a beam of light illuminating his face.", "The image shows a person sitting on the chair. His appearance is barely recognizable in the dim environment." ], "emotion": [ "The image shows a person, with a big smile, sitting on the chair", "The image shows an angry person sitting on the chair" ], "posture": [ "The image shows a person sitting relaxing on the chair.", "The image shows a person standing straight in fromt of the chair." ], "size & shape": [ "The image shows a person sitting on a round, cylindrical chair.", "The image shows a person sitting on a square-shaped chair." ] } }

Original Caption: The image shows an apple on the table.
Enriched Captions:
{ "captions": { "color": [ "The image shows a red apple on the table.", "The image shows a green apple on the table." ], "size & shape": [ "The image shows a big ripe apple on the table.", "The image shows a rotten apple on the table." ] } }

Original Caption: The image illustrates an air balloon.
Enriched Captions:
{ "captions": { "light condition": [ "The image illustrates an air balloon in a dark room.", "The image illustrates an air balloon in a bright room.", ], "size & shape": [ "The image illustrates a deflated air balloon.", "The image illustrates an inflated air balloon." ], "color": [ "The image illustrates a light blue air balloon.", "The image illustrates an air balloon in yellow color." ] } }

Now please generate enriched captions based on the following original caption. Ensure your response follows the JSON format of the above examples.
Original Caption:
[original_caption]
Enriched Captions:

Table 18: The prompt used to generate textual temporal QA for the *Attribute* aspect.

You will be presented with several pairs of image captions. Each pair of captions depicts two keyframes in a video. Your task is to generate multi-choice questions (and corresponding answers) for each pair of captions. The questions should focus on the change of attribute between the keyframe contents. Ensure that the questions are diverse and distinct from each other in wordings.

Here are some examples of caption pairs and generated question-answer pairs:

Caption Pair 1:
1. The image shows a person sitting on the chair with a beam of light illuminating his face.
2. The image shows a person sitting on the chair. His appearance is barely recognizable in the dim environment.

Caption Pair 2:
1. The image shows a person, with a big smile, sitting on the chair.
2. The image shows an angry person sitting on the chair.

Questions and Answers:
{ "qas": { "caption_pair_1": { "question": "How does the light condition change in the video?", "options": [ "remaining stable", "turning darker", "turning brighter" ], "answer": "turning darker" }, "caption_pair_2": { "question": "What change occurs to the person in the video?", "options": [ "changing from smiling to being angry", "changing from being angry to smiling", "changing from feeling shy to being angry", "changing from feeling awkward to smiling" ], "answer": "changing from smiling to being angry" } } }

Caption Pair 1:
1. The image illustrates an air balloon in a dark room.
2. The image illustrates an air balloon in a bright room.

Caption Pair 2:
1. The image illustrates a deflated air balloon.
2. The image illustrates an inflated air balloon.

Caption Pair 3:
1. The image illustrates a light blue air balloon.
2. The image illustrates an air balloon in yellow color.

Questions and Answers:
{ "qas": { "caption_pair_1": { "question": "What transformation is occurring in the brightness of the video?", "options": [ "increasing", "staying the same", "decreasing" ], "answer": "increasing" }, "caption_pair_2": { "question": "What is happening to the shape of the air balloon?", "options": [ "it is getting bigger", "it is getting smaller", "its size and shape remains consistent" ], "answer": "it is getting bigger" }, "caption_pair_3": { "question": "How can we describe the change happening to the air balloon?", "options": [ "its color changes from grey to yellow", "its color changes from light blue to yellow", "its color changes from yellow to light blue", "its color changes from yellow to green" ], "answer": "its color changes from light blue to yellow" } } }

Now please generate question-answer pairs based on the following caption pairs. Ensure your response follows the above JSON format and style of the example question-answer pairs.
**[caption_pairs]**
Questions and Answers:

Table 19: The prompt used to generate similar captions for the *Temporal Referring* aspect.

You will be presented with an original image caption. Your task is to modify this caption by changing the original elements including object, action and attribute. Ensure that the modified element is distinct from the original ones.

Here is an example of original caption and modified captions:

Original Caption: The image shows a person sitting on the chair.
Modified Captions:
{ "captions": { "change_object": [ "The image shows a cat sitting on the chair.", "The image shows a dog sitting on the chair.", "The image shows a cup placed on the chair." ], "change_action": [ "The image shows a person standing next to the chair.", "The image shows a person sleeping on the chair.", "The image shows a person dancing nearby the chair." ], "change_attribute": [ "The image shows a tall person sitting on the chair.", "The image shows a short person sitting on the chair.", "The image shows a strong person sitting on the chair." ] } }

Now please generate modified captions based on the following original caption. Ensure your response follows the JSON format of the above example.
Original Caption:
[original_caption]

Table 20: The prompt used to generate declarative statements for the *Temporal Grounding* aspect.

You will be given a question paired with several answers. Your task is to reformulate each answer into a simple declarative statement.

###Example1
Question: What is the color of the cat?
Answer 1: white
Answer 2: orange
Answer 3: black
Declarative Statement 1: the cat is white
Declarative Statement 2: the cat is orange
Declarative Statement 3: the cat is black

###Example2
Question: What is the person doing?
Answer 1: running
Answer 2: playing guitar
Answer 3: swimming
Declarative Statement 1: the person is running
Declarative Statement 2: the person is playing guitar
Declarative Statement 3: the person is swimming

Now please reformulate the following question and answers according to the above examples.
[question_and_answers]

Table 21: Datasets used in our textual temporal transfer explorations in Table 2.

| Dataset | # samples | Description |
|---|---|---|
| Open-LLaVA-Next | 22k | LLaVA-Next image-text SFT dataset. |
| HotpotQA | 22k | Multiple-document question answering dataset. |
| Temporal Change (w/o Distractor) | 22k | Balanced mixture of Order-GPT and Attribute without interesting distractor captions |
| Temporal Change (1x) | 22k | Balanced mixture of Order-GPT (1x) and Attribute (1x) |
| Temporal Change (2x) | 22k | Balanced mixture of Order-GPT (2x) and Attribute (2x) |
| Temporal Change (4x) | 22k | Balanced mixture of Order-GPT (4x) and Attribute (4x) |
| Temporal Change (8x) | 22k | Balanced mixture of Order-GPT (8x) and Attribute (8x) |
| Temporal Change (1-8x) | 22k | Balanced mixture of Temporal Change 1x, 2x, 4x and 8x |
| Order-Template | 22k | Balanced Mixture of Order-Template (phrase), (prefix) and (sentence) |
| Temporal Referring | 22k | Synthesized textual samples for enhancing Referring |
| Temporal Grounding | 22k | Synthesized textual samples for enhancing Grounding |
| Order-Template + Temporal Change (1x) | 22k | Balanced mixture of Order-Template and Temporal Change (1x) |
| Temporal Grounding + Temporal Change (1x) | 22k | Balanced mixture of Temporal Grounding and Temporal Change (1x) |
| Temporal Referring + Temporal Change (1x) | 22k | Balanced Mixture of Temporal Referring and Temporal Change (1x) |
| T3 | 22k | Balanced Mixture of all our synthesized subtasks. |

Table 22: Datasets used for transferring to holistic video understanding benchmarks in Table 3 and Table 4.

| Dataset | # samples | Description |
|---|---|---|
| LLaVA-Next | 200k | LLaVA-Next image-text SFT dataset. |
| HotpotQA w/ LLaVA-Next | 200k | Hotpot QA (66.7k) + + LLaVA-Next (133.3k) |
| T3 w/ LLaVA-Next (Ours) | 200k | Temporal All (66.7k) + LLaVA-Next (133.3k) |

Table 23: Performance continually increases with more training samples on Video-MME and MLVU benchmarks.

| Model | Video-MME (Short / Medium / Long / Overall) | MLVU (Macro-Avg) |
|---|---|---|
| LongVA | 61.1 / 50.4 / 46.2 / 52.6 | 56.4 |
| + 100K T3 w/ LLaVA-Next | 61.7 / 51.0 / 45.7 / 52.8 | 56.9 |
| + 200K T3 w/ LLaVA-Next | **63.3 / 54.8 / 46.8 / 55.0** | **58.1** |

Table 24: Qwen2-VL-72B results on MLVU with different frame settings.

| MLVU | Action Count | Ego Reasoning | Needle QA | Action Order | Plot QA | Anomaly Recognition | Topic Reasoning | Macro-Average |
|---|---|---|---|---|---|---|---|---|
| Qwen2-VL-72B (16 frame) | 20.4 | 52.3 | 55.8 | 46.7 | 62.0 | 65.5 | 85.6 | 55.5 |
| + T3 w/ LLaVA-Video (Ours) | 31.6 (+11.2) | 55.7 (+3.4) | 67.0 (+11.2) | 54.1 (+7.4) | 62.0 | 61.0 (-4.5) | 86.3 (+0.7) | 59.7 (+4.2) |
| Qwen2-VL-72B (32 frame) | 22.8 | 54.8 | 65.1 | 46.3 | 63.6 | 71.0 | 87.5 | 58.7 |
| + T3 w/ LLaVA-Video (Ours) | 32.0 (+9.2) | 59.9 (+5.1) | 70.4 (+5.3) | 54.4 (+8.1) | 64.0 (+0.4) | 69.0 (-2.0) | 87.8 (+0.3) | 62.5 (+3.8) |

Table 25: Qwen2-VL-72B results on Video-MME with different frame settings.

| Video-MME | Short | Medium | Long | Overall |
|---|---|---|---|---|
| Qwen2-VL-72B (16 frame) | 72.3 | 60.2 | 55.8 | 62.8 |
| + T3 w/ LLaVA-Video (Ours) | 73.8 (+1.5) | 62.6 (+2.4) | 54.9 (-0.9) | 63.7 (+0.9) |
| Qwen2-VL-72B (32 frame) | 75.8 | 62.9 | 56.6 | 65.1 |
| + T3 w/ LLaVA-Video (Ours) | 78.3 (+2.5) | 64.3 (+1.4) | 57.1 (+0.5) | 66.6 (+1.5) |

Table 26: Percentage of T3 validation set examples that are similar (>0.5 Jaccard Similarity) to any examples in the video benchmark, in terms of Question and Choice. The results are averaged across five times of random sampling with the standard deviation reported in the parentheses.

| | Question | | | Choice | | |
|---|---|---|---|---|---|---|
| | TempCompass | Video-MME | MLVU | TempCompass | Video-MME | MLVU |
| Attribute | 0.0(0.0) | 0.0(0.0) | 0.04(0.08) | 1.5(0.4) | 0.1(0.1) | 2.7(0.5) |
| Order-Template (phrase) | 0.0(0.0) | 0.0(0.0) | 0.0(0.0) | 0.0(0.0) | 0.0(0.0) | 0.0(0.0) |
| Order-Template (prefix) | 0.0(0.0) | 0.0(0.0) | 0.0(0.0) | 0.0(0.0) | 0.0(0.0) | 0.0(0.0) |
| Order-Template (sentence) | 0.0(0.0) | 0.0(0.0) | 0.0(0.0) | - | - | - |
| Order | 0.0(0.0) | 0.8(0.2) | 0.3(0.1) | 0.7(0.4) | 0.8(0.2) | 0.3(0.1) |
| Grounding | 0.0(0.0) | 0.0(0.0) | 0.0(0.0) | 0.7(1.4) | 0.8(1.6) | 0.0(0.0) |
| Referring | 0.0(0.0) | 0.0(0.0) | 0.2(0.2) | 2.7(0.4) | 0.5(0.2) | 4.2(0.5) |

Table 27: Comparison results of video inputs effect on TempCompass.

| TempCompass | Action | Direction | Speed | Order | Attribute_change | Average |
|---|---|---|---|---|---|---|
| LongVA-7B w/ video input | 92.3 | 37.3 | 42.0 | 54.3 | 51.7 | 55.9 |
| LongVA-7B w/o video input | 50.3 | 36.1 | 42.0 | 39.7 | 38.9 | 41.5 |
| T3 (Ours) w/o video input | 48.8 | 36.7 | 44.5 | 38.4 | 37.2 | 41.3 |

Table 28: Comparison results of video inputs effect on Video-MME.

| Video MME | Short | Medium | Long | Overall |
|---|---|---|---|---|
| LongVA-7B w/ video input | 61.1 | 50.4 | 46.2 | 52.6 |
| LongVA-7B w/o video input | 34.9 | 35.2 | 37.6 | 35.9 |
| T3 (Ours) w/o video input | 34.8 | 35.3 | 34.4 | 34.9 |

Table 29: Comparison results of video inputs effect on MLVU.

| MLVU | Action Count | Ego Reasoning | Needle QA | Action Order | Plot QA | Anomaly Recognition | Topic Reasoning | Macro-Average |
|---|---|---|---|---|---|---|---|---|
| LongVA-7B w/ video input | 25.2 | 48.6 | 70.4 | 41.7 | 68.1 | 58.5 | 82.2 | 56.4 |
| LongVA-7B w/o video input | 33.0 | 34.4 | 50.7 | 36.3 | 43.2 | 28.0 | 36.7 | 37.5 |
| T3 (Ours) w/o video input | 32.5 | 34.1 | 49.3 | 42.9 | 41.2 | 25.5 | 29.2 | 36.4 |

