# OpenReview forum: "Temporal Reasoning Transfer from Text to Video"
_ICLR.cc/2025/Conference — ICLR 2025 Poster_

### Official Review · Reviewer_c8nY · 2024-10-31

**Soundness:** 3
**Presentation:** 4
**Contribution:** 3
**Rating:** 8
**Confidence:** 4

**Summary:**

This paper addresses the limitations in Vision Language Models (VLMs) for temporal reasoning tasks in video understanding. Through detailed probing experiments, the authors identify the primary bottleneck as the LLM component rather than the visual encoder. To address this, they introduce an approach called T3, which fine-tunes only the LLM component on text-only data, enhancing performance on several temporal reasoning benchmarks.

**Strengths:**

1. The authors provide a well-defined motivation for their work, verifying through experiments that the LLM, rather than the VLM, is the limiting factor in video understanding tasks, and proposed solutions.
2. The authors introduce an innovative fine-tuning technique that only tunes the LLM within a VLM, bypassing the need for visual data. This approach yields significant performance improvements across several video understanding benchmarks.
3. The dataset is generated in a fully automated manner, adding to the scalability of their approach.
4. Extensive experiments and in-depth analysis are provided.

**Weaknesses:**

1. As mentioned in L-538, the approach shows minimal improvement when applied to larger LLMs, which could restrict its broader applicability.
2. Several areas of the paper would benefit from further clarification (see questions below).

**Questions:**

1. Could you report results on larger LLMs (e.g., Qwen2-72B)? Although the performance gain is noted to be limited, these results could be informative for future research.
2. Could you provide examples of captions generated by GPT-4o as mentioned in L-133? It would help clarify whether specific attribute information is inadequately encoded in the caption, potentially leading to near-random results in LLM responses. (You partial addressed this in L216-L217, but the performance achieved by larger model is still 20% percent lower than the performance achieved by LSTM, which makes it not convincing).
3. Minor issue: It appears LLaVA-ReCap-558K is derived from LCS-558K (a subset of the LAION/CC/SBU dataset) rather than BLIP558K (as stated in lines 861-862) [1].
4. I question whether the performance improvement is due to enhanced instruction-following ability (e.g., the ability to "Answer the option only") rather than true temporal reasoning. Could you report the portion of mis-formatted or invalid responses across all experiments to clarify this?

[1] Section on Open Source Release: https://llava-vl.github.io/blog/2024-05-25-llava-next-ablations/

---

> ### Author Response · Authors · 2024-11-22
> **Response to Reviewer c8nY (Part 1)**
>
> Dear Reviewer c8nY,
>
> We are greatly encouraged by your recognition of our `well-defined motivation`, the `innovative fine-tuning technique that bypasses the need for visual data`, and our `thorough experimental analysis across multiple benchmarks`.
>
> We address the questions you pointed out as follows:
> ### **1. Examples of captions generated by GPT-4o**
> Thanks for raising this important question! We have demonstrated the GPT-4o generated frame captions in Tables 9, 10, and 11 in the appendix. Below are two examples of captions for the temporal aspects of brightness change and blooming.
>
> **Brightness Change**
>
> - Frame 1: The image shows a **very dark** scene with a black and white cat lying on top of a laptop keyboard. The cat’s face and whiskers are slightly visible, but the overall image is hard to see due to the low brightness.
> - Frame 3: The image shows a black and white cat lying on top of a laptop keyboard. The overall scene is **very dark**, making it difficult to see details clearly.
> - Frame 5: The image shows a black and white cat lying on top of a laptop keyboard in a **slightly dark** environment.
> - Frame 7: The image shows a black and white cat lying on top of a laptop keyboard. The cat appears to be comfortably settled in the space between the laptop screen and keyboard. The overall **brightness of the image is normal**, allowing clear visibility of the cat’s fur and the laptop’s keys.
>
> **Blooming**
>
> - Frame 1: This frame shows a blue **bud** at the end of a green cotton swab positioned vertically against a plain white background. The cotton bud appears clean and unused, with no visible movement or action taking place.
> - Frame 3: This frame shows the same blue **bud** at the end of a green cotton swab, still positioned vertically against a plain white background. The cotton on the bud appears slightly more compressed and misshapen compared to the preceding frame.
> - Frame 5: This frame shows a blue **flower with five petals** on a green stem, set against a white background. The flower appears to be blooming mid-stage with slightly spread petals, and the stem is straight and upright, positioned centrally in the frame.
> - Frame 7: This frame features the same blue **flower with six petals** on a green stem, now slightly to the right against a white background. The petals are still widely spread, and the stem remains upright, suggesting no noticeable movement or change from the previous frame.
>
> As we can see, these captions provide sufficient and accurate per-frame information to comprehend the temporal dynamics of brightness change and blooming. We will update the paper to make references to Tables 9, 10, and 11 clearer in the main text. Thanks again for the constructive advice.
>
> ### **2. LLaVA-ReCap-558K is derived from LCS-558K rather than BLIP558K**
> Thank you for bringing this issue to our attention. Our information was initially based on Section 3.2 of the blog you referenced (https://llava-vl.github.io/blog/2024-05-25-llava-next-ablations/), which stated, "We used the model to generate new captions for the images from the following datasets: COCO118K, BLIP558K, and CC3M." However, it appears that "LCS-558K (a subset of the LAION/CC/SBU dataset)" is a more accurate description of the original source of the images. We will since update our paper to reflect this information accurately. Thank you again for this careful suggestion.

---

> > ### Author Response · Authors · 2024-11-22
> > **Response to Reviewer c8nY (Part 2)**
> >
> > ### **3. Is the performance improvement due to enhanced instruction-following ability?**
> > Great question! Our evaluation of the model response follows the default setting of lmms-eval (https://github.com/EvolvingLMMs-Lab/lmms-eval).
> >
> > For MLVU and our text validation sets, we use GPT-4o-mini to compare the Video LLM responses with the ground-truth answers. Mis-formatted responses are not considered incorrect as long as their semantic content aligns with the ground-truth.
> >
> > For Video-MME and TempCompass, the percentage of mis-formatted responses is reported in the tables below. As we can see, the original LongVA-7B already produces responses in the expected format for all benchmark examples, indicating that **the improvement of our method is not due to enhanced instruction-following ability.**
> >
> >
> > | Video-MME  |  Mis-Formatted Response  |
> > |:------:|:-----:|
> > | LongVA-7B      |  0.00% |
> > |+ 100K T3 w/ LLaVA-Next    |  0.00% |
> > |+ 200K T3 w/ LLaVA-Next    |  0.04% |
> >
> > | TempCompass  |  Mis-Formatted Response  |
> > |:------:|:-----:|
> > |LongVA-7B      |  0.00% |
> > |+ 100K T3 w/ LLaVA-Next    |  0.95% |
> > |+ 200K T3 w/ LLaVA-Next    |  4.87% |
> >
> >  Furthermore, Table 5 in the Appendix presents an ablation study where images replace their corresponding captions while maintaining all other conditions (same questions and answers). This controlled setting also examines instruction-following effects. Image-based transfer only marginally improves Order performance from 54.3 to 55.4, while textual format transfer significantly boosts it to 63.6. This substantial gap further confirms that the improvement stems from enhanced temporal reasoning capabilities rather than better instruction-following.
> >
> >
> > ### **4. Results on larger LLMs**
> > Thank you for the constructive feedback. We are currently training larger models, i.e., Qwen2-VL-72B, with our T3 dataset. Should the results become available before the rebuttal deadline, we will notify you.

---

> ### Comment · Reviewer_c8nY · 2024-11-22
>
> Thank you for your response. Most of my concerns are addressed so I increased my ratings (5->6). I'm looking forward to your results on at least one of larger models.

---

> > ### Author Response · Authors · 2024-11-26
> > **Thank you**
> >
> > We sincerely thank you for the constructive comments and we are glad to see our rebuttal addressed your concerns.
> >
> > Following your suggestion, we have evaluated our method on Qwen2-VL-72B, one of the latest large vision-language models (detailed results in our Response to Reviewer UiVz (Part I)).
> > T3 consistently improves performance even for models up to 72B, with significant boosts on key temporal reasoning tasks (e.g., Action Count: +9.2 and Action Order: +8.1 with 32 frames on MLVU).
> >
> > We will incorporate these new results in our revision to strengthen our manuscript.
> >
> > Thank you again for the valuable feedback!

---

> > > ### Comment · Reviewer_c8nY · 2024-11-26
> > >
> > > Thank you for running such large experiments. The results look fantastic. Therefore I further increased my rating to 8.

---

### Official Review · Reviewer_SHtr · 2024-11-02

**Soundness:** 2
**Presentation:** 3
**Contribution:** 2
**Rating:** 6
**Confidence:** 4

**Summary:**

The paper studies the video's large language models, particularly its temporal reasoning ability. It first designs probing experiments using synthesized videos and analyzes the bottleneck in temporal understanding: primarily on LLMs but not the vision encoder.  To mitigate the issue, the authors propose a simple text data augmentation method to leverage the image-text datasets to generate diverse
temporal reasoning tasks and enhance the video LLMs training. The experiments based on LongVA-7B show its competitive performances across different datasets.

**Strengths:**

1. The paper is well structured and written, making it easy to understand its key points.

2. The enhanced model, built on LongVA-7B, has shown strong performances, surpassing large-scale models such as InternVLChat-V1.5-20B and VILA1.5-40B.

**Weaknesses:**

1. The claims of the paper do not fully convince me of the paper made. I agree that most of the vision encoders used today do not help much with temporal reasoning, even though they are trained with millions of videos. However, this can not be claimed as "the temporal reasoning most of rely on the temporal reasoning ability from LLMs. There might be reasons that the temporal reasoning also replies on the model alignment ability between the text and the images/videos. The authors should design extensive prompting experiments on more models and datasets to support their strong claims.

2. The synthesized tasks are mainly based on GPT-4o. And for me, how to design the synthesized tasks seems very arbitrary and empirical. Are there any particular reasons you designed these tasks and how can you ensure its generalization on more domain/datastes? The authors should have more deep analysis and experiments on those two issues.

**Questions:**

Besides the questions I listed before, there are two additional questions for the authors:

1. How to explain the strong temporal reasoning ability of GPT-4o? Is it only because 4o has a very strong temporal reasoning ability from the language side, not related to its vision encoder or alignment?

2. Is there any way you can conduct the probing experiments on real datasets, not the synthesized ones?

---

> ### Author Response · Authors · 2024-11-24
> **Response to Reviewer SHtr (Part 1)**
>
> Dear Reviewer SHtr,
>
> We greatly appreciate your thoughtful feedback on our work! We are encouraged by your recognition of our `clear writing` and `the effectiveness of our method`. We address your questions and concerns as follows:
>
> ### **Weakness 1. Is LLM the major bottleneck in video temporal reasoning?**
> Thanks for raising this important question. Indeed, the vision encoder, vision-language alignment and the LLM are all essential for video temporal reasoning. However, in our work, we argue that while the vision encoder and alignment components already provide sufficient visual information, the LLM remains the major bottleneck due to its insufficient temporal reasoning ability. This claim is supported by the following evidence:
>
> - Even when provided with per-frame textual captions as context, 7B-scale LLMs (e.g., Qwen2 and LLaMA3) fail to correctly answer a significant portion of questions in basic temporal probing tasks. This highlights a critical limitation in their ability to handle temporal reasoning.
> - Classifier probes trained on **visual features projected into the LLM embedding space** achieve over 90% accuracy in most temporal aspects. This demonstrates the high quality of both the vision encoder and vision-language alignment components, further isolating the LLM as the bottleneck.
> - Our study involves two Video LLMs, LongVA-7B and VILA-8B, built on Qwen2-7B and LLaMA3-8B, respectively. These models exhibit similar trends in our probing study, reinforcing the robustness of our conclusion.
> - TempCompass [1] reveals that while advanced Video LLMs achieve high accuracy in recognizing coarse-grained and fine-grained actions, their performance is substantially weaker in temporally reliant dimensions such as event order, attribute changes, and directional reasoning. This finding aligns with our claim that the LLM’s temporal reasoning ability is the major bottleneck.
> - In TOMATO [2], a recently proposed benchmark for video temporal understanding, it was observed (in Section 6.3) that “*GPT-4o correctly generates captions for each consecutive change in the moon’s movement, showcasing its ability to reason at individual time steps. However, it fails to infer based on the captions that the overall sequence represents a clockwise rotation.*” This finding suggests that vision-language alignment is not a significant bottleneck for the most advanced Video LLMs.
>
> In summary, our claim that “LLM is the major bottleneck of video temporal reasoning” is well-supported by our rigorous empirical results & previous/concurrent studies.
>
> [1] TempCompass: Do Video LLMs Really Understand Videos?
>
> [2] TOMATO: Assessing Visual Temporal Reasoning Capabilities in Multimodal Foundation Models.

---

> > ### Author Response · Authors · 2024-11-24
> > **Response to Reviewer SHtr (Part 2)**
> >
> > ### **Weakness 2. The reason we design the textual temporal tasks and whether they can generalize to more domains/datasets?**
> > Great question! Our textual temporal tasks focus on four temporal aspects, i.e., event order, attribute change, temporal referring and temporal grounding, which is inspired by recent video understanding benchmarks [1,2,3,4]. We choose these four particular aspects because: (1) They are very basic and serve as the foundation for more complex temporal reasoning skills. (2) Our probing study in Section 2 reveals that the two 7B-scale Video LLMs, i.e., LongVA-7B and VILA-8B, cannot fully understand these basic temporal aspects.
> >
> > To assess whether our method can generalize to more temporal aspects, domains and datasets, we evaluated the models on two recently released benchmarks: TOMATO [5] and Vinoground [6].
> >
> > TOMATO tests a variety of motion-related temporal reasoning abilities, including direction, velocity, and rotation. The results below demonstrate that our method significantly improves performance in velocity and frequency understanding (39.9% vs 23.1% baseline) and Shape & Trend (30.5% vs 22.4%). While the performance in Rotation still leaves room for improvement, the findings demonstrate that **our textual temporal tasks can generalize effectively to unseen motion-related temporal concepts encountered during evaluation.**
> >
> > **Results on TOMATO**
> > | **Model**            | **Rotation** | **Direction** | **Velocity & Frequency** | **Shape & Trend** | **Visual Cues** | **Action Count** | **All** |
> > | -------------------- | ------------ | ------------- | ------------------------ | ----------------- | --------------- | ---------------- | ------- |
> > | GPT-4o               | 24.5         | 45.2          | 31.9                     | 42.6              | 58.6            | 36               | 37.7    |
> > | LLaVA-NeXT-Video-32B | 20.6         | 26.3          | 12.4                     | 24.2              | 30              | 24.3             | 22.7    |
> > | VideoLLaMA2 72B      | 14.3         | 24.6          | 22.4                     | 26.5              | 27.1            | 28.8             | 23.5    |
> > | LongVA-7B            | 26.4         | 21.6          | 23.1                     | 22.4              | 38.6            | 14.7             | 22.3    |
> > | + LLaVA-Next 200k    | 22.2         | 25.6          | 21.2                     | 26.0              | 40.0            | 19.8             | 21.2    |
> > | + T3 (Ours) 200k     | 16.6         | 21.6          | 39.9                     | 30.5              | 38.6            | 16.4             | 24.3    |
> >
> > Vinoground comprises carefully curated real-world video-caption pairs, with each pair sharing identical static content but differing in subtle temporal nuances. For example, "*the watermelon is cut and then turned*" versus "*the watermelon is turned and then cut*" is a pair of positive/negative captions. In this benchmark, models are challenged to identify the correct caption given a video (Text) or the correct video given a caption (Video).
> > As shown in the results below, our T3 dataset significantly enhances the performance of LongVA-7B on Vinoground. Furthermore, our approach demonstrates noticeable improvements in tasks such as Action Counting (+3.9) and Plot QA (+1.3) on MLVU, even though these tasks are not present in the T3 dataset. These findings highlight that **our temporal textual tasks generalize effectively to various downstream domains and datasets.**
> >
> > **Results on Vinoground**
> > | Model                | Text          | Video        | Group       |
> > | -------------------- | ------------- | ------------ | ----------- |
> > | GPT-4o               | 54.0          | 38.2         | 24.6        |
> > | LLaVA-Next-Video 34B | 23.00         | 21.2         | 3.8         |
> > | LLaVA-NeXT-Video-7B  | 21.8          | 25.6         | 6.2         |
> > | LongVA-7B            | 21.2          | 21.6         | 5.4         |
> > | + 200k LLaVA-Next    | 19.0 (- 2.2)  | 21.8 (+0.2)  | 4.6 (-0.8)  |
> > | + 200k T3 (Ours)     | 31.6 (+ 10.4) | 30.0 ( +8.4) | 11.8 (+6.4) |
> >
> > [1] TempCompass: Do Video LLMs Really Understand Videos?
> >
> > [2] Video-MME: The First-Ever Comprehensive Evaluation Benchmark of Multi-modal LLMs in Video Analysis.
> >
> > [3] Vitatecs: A diagnostic dataset for temporal concept understanding of video-language models.
> >
> > [4] Longvideobench: A benchmark for long-context interleaved video-language understanding.
> >
> > [5] TOMATO: Assessing Visual Temporal Reasoning Capabilities in Multimodal Foundation Models.
> >
> > [6] Vinoground: Scrutinizing LMMs over Dense Temporal Reasoning with Short Videos.

---

> > > ### Author Response · Authors · 2024-11-24
> > > **Response to Reviewer SHtr (Part 3)**
> > >
> > > ### **Question 1. How to explain the strong temporal reasoning ability of GPT-4o?**
> > > Unfortunately, we cannot provide a precise answer, as the technical details of GPT-4o have not been publicly disclosed. However, we can make an educated guess that GPT-4o incorporates all three essential elements: a robust vision encoder, a well-aligned vision-language space, and a powerful language model. Hypothetically, it is built upon a large-scale language model (possibly exceeding 70 billion parameters) and trained using high-quality multimodal datasets, including tasks focused on temporal reasoning.
> > >
> > > ### **Question 2. Is there any way you can conduct the probing experiments on real datasets, not the synthesized ones?**
> > > Conducting probing experiments on existing real datasets is not feasible. This is because probing visual features requires collecting videos for distinct temporal categories (e.g., cat->person versus person->cat). To the best of our knowledge, no existing real video dataset provides the necessary categorization for these temporal distinctions. Additionally, it is important to note that the two Video LLMs (LongVA-7B and VILA-8B) and their corresponding LLMs (Qwen2-7B and LLaMA3-8B) perform poorly even on relatively simple synthesized probing tasks. This indicates that their performance would likely be even worse on real video data.

---

> > > > ### Comment · Reviewer_SHtr · 2024-11-27
> > > > **Comments from Reviewer SHtr**
> > > >
> > > > Thank you for the authors' rebuttal, which addressed some of my main concerns. I increase my score from 5 to 6.

---

> > > > > ### Author Response · Authors · 2024-11-28
> > > > > **Thank you**
> > > > >
> > > > > Dear Reviewer SHtr,
> > > > >
> > > > > We are pleased that our rebuttal effectively addressed your concerns and deeply grateful for your decision to raise the score of our paper.
> > > > >
> > > > > Thank you once more for your valuable time and insightful feedback!

---

### Official Review · Reviewer_hvnd · 2024-11-03

**Soundness:** 3
**Presentation:** 3
**Contribution:** 3
**Rating:** 6
**Confidence:** 4

**Summary:**

The paper attempts to address current limitations in temporal reasoning of Video Large Language Models (Video LLMs). Current models still often struggle with understanding temporal sequences, despite advances in video comprehension. Through a detailed analysis, the authors empirically demonstrate that the bottleneck primarily lies in the LLM's capability to understand temporal concepts. This is in contrast to a multitude of concurrent work that often aim to improve the temporal encoding or adaptor functions. To address this limitation, this paper also introduces the Textual Temporal Reasoning Transfer (T3) approach, that generates temporal reasoning tasks solely in textual format. The experiments reported in the paper demonstrates the effectiveness of the approach where it obtains performance gains in multiple video question answering benchmarks, without requiring additional video and language annotations for training.

**Strengths:**

1) In terms of significance, this paper addresses the problem of video and language reasoning in multimodal Large Language Models (MLLMs) from a new perspective. It attributes the limitation of existing (MLLMs) to the LLM component instead of the video or adaptor functions. This is a very interesting perspective since it proposes an approach that improves the LM's ability to understand temporal concepts through text. The proposed Textual Temporal Reasoning Transfer (T3) approach creatively uses text-only datasets to improve temporal reasoning without requiring any video-language annotations.

2) The quality of the paper is relatively high, especially with regards to the empirical analysis of the bottleneck in video and language reasoning. The experiments conducted in the analysis are well motivated. The finding that the performance of existing MLLMs is constrained by their LLM decoders instead of the visual encoding function is insightful.

3) In terms of clarity, the paper is well-written and motivated. The model figures are informative and especially helpful in helping the reader to understand the differences between questions that focus on different temporal reasoning abilities in videos.

**Weaknesses:**

1) The text-based tasks used in T3 are largely devised based on templates and may not always be similar to the language concepts that are contained in real-world videos. For example, such templated sentences may lack the subtle temporal cues that naturally occur in descriptions of complex events. This might affect the model's ability to generalize to different downstream tasks.

2) In this work, the authors only address four temporal aspects: order, attribute change, temporal referring, and grounding. However, this excludes some other common aspects of temporal reasoning such as understanding causal relationships and durations. Including more training samples that require more complex forms of reasoning could improve robustness of the trained models across a broader spectrum of temporal tasks.

**Questions:**

See weaknesses.

---

> ### Author Response · Authors · 2024-11-23
> **Response to Reviewer hvnd (Part 1)**
>
> Dear Reviewer hvnd,
>
> Thank you for the thoughtful review of our work on improving temporal reasoning in Video LLMs. We sincerely appreciate your recognition of the `novel perspective`, the `quality of our empirical analysis`, and the `clarity of our writing`.  Our responses to your questions are below.
>
> ### **Weakness 1: Can our method generalize to downstream tasks with subtle temporal cues?**
> We fully understand your concern that our template-based text tasks may not generalize effectively to real-world videos with subtle temporal cues. To address this, we evaluated the models on the recently released Vinoground benchmark [1]. Vinoground comprises carefully curated real-world video-caption pairs, with each pair sharing identical static content but differing in subtle temporal nuances. For example, "*the watermelon is cut and then turned*" versus "*the watermelon is turned and then cut*" is a pair of positive/negative captions. In this benchmark, models are challenged to identify the correct caption given a video (Text) or the correct video given a caption (Video).
>
> As shown in the results below, our T3 dataset significantly enhances the performance of LongVA-7B on Vinoground. Furthermore, our approach demonstrates noticeable improvements in tasks such as Action Counting (+3.9) and Plot QA (+1.3) on MLVU, even though these tasks are not present in the T3 dataset. These findings highlight that **while our method relies on template-based text tasks, it generalizes effectively to downstream tasks involving real-world videos and subtle temporal cues.**
>
> | Model                | Text          | Video        | Group       |
> | -------------------- | ------------- | ------------ | ----------- |
> | GPT-4o               | 54.0          | 38.2         | 24.6        |
> | LLaVA-Next-Video 34B | 23.00         | 21.2         | 3.8         |
> | LLaVA-NeXT-Video-7B  | 21.8          | 25.6         | 6.2         |
> | LongVA-7B            | 21.2          | 21.6         | 5.4         |
> | + 200k LLaVA-Next    | 19.0 (- 2.2)  | 21.8 (+0.2)  | 4.6 (-0.8)  |
> | + 200k T3 (Ours)     | 31.6 (+ 10.4) | 30.0 ( +8.4) | 11.8 (+6.4) |
>
> Furthermore, we have identified two promising directions to enhance our approach to better address the subtle temporal cues in real-world videos:
> - **Diversifying text formats by leveraging MLLMs to automatically derive natural temporal descriptions from real videos and images.** This would help capture more organic temporal expressions and relationships beyond templated patterns.
>
> - **Mixing modalities by combining text with images/videos to better capture subtle temporal nuances.** This multimodal approach could help bridge the gap between templated and natural temporal expressions.
>
> We would like to explore these ideas in the future and incorporate the discussion into our revision.
>
> [1] Vinoground: Scrutinizing LMMs over Dense Temporal Reasoning with Short Videos.

---

> ### Author Response · Authors · 2024-11-23
> **Response to Reviewer hvnd (Part 2)**
>
> ### **Weakness 2: Regarding a broader spectrum of temporal tasks.**
> Thanks for the constructive feedback! We agree that incorporating more diverse and complex forms of temporal reasoning could enhance the robustness of our method, and we look forward to exploring this in future work. However, we would like to emphasize that the four temporal aspects in our paper are very basic and serve as the foundation for more complex temporal reasoning skills.
>
> To validate this hypothesis, we evaluated our method on the newly released TOMATO benchmark [1], which tests a variety of motion-related temporal reasoning abilities, including direction, velocity, and rotation.
>
> Our findings reveal promising generalization to these temporal concepts unseen in our T3 training dataset: For instance, our method significantly improves performance in velocity and frequency understanding (39.9% vs 23.1% baseline) and Shape & Trend (30.5% vs 22.4%). While there is room for improvement in categories like Rotation, we plan to explore enhanced approaches using the directions discussed above. These new results will be included in our revision.
>
> | **Model**            | **Rotation** | **Direction** | **Velocity & Frequency** | **Shape & Trend** | **Visual Cues** | **Action Count** | **All** |
> | -------------------- | ------------ | ------------- | ------------------------ | ----------------- | --------------- | ---------------- | ------- |
> | GPT-4o               | 24.5         | 45.2          | 31.9                     | 42.6              | 58.6            | 36               | 37.7    |
> | LLaVA-NeXT-Video-32B | 20.6         | 26.3          | 12.4                     | 24.2              | 30              | 24.3             | 22.7    |
> | VideoLLaMA2 72B      | 14.3         | 24.6          | 22.4                     | 26.5              | 27.1            | 28.8             | 23.5    |
> | LongVA-7B            | 26.4         | 21.6          | 23.1                     | 22.4              | 38.6            | 14.7             | 22.3    |
> | + LLaVA-Next 200k    | 22.2         | 25.6          | 21.2                     | 26.0              | 40.0            | 19.8             | 21.2    |
> | + T3 (Ours) 200k     | 16.6         | 21.6          | 39.9                     | 30.5              | 38.6            | 16.4             | 24.3    |
>
> [1] TOMATO: Assessing Visual Temporal Reasoning Capabilities in Multimodal Foundation Models

---

> > ### Comment · Reviewer_hvnd · 2024-11-26
> > **Response to authors**
> >
> > Thank you very much for your detailed responses! In particular, I would like to express my appreciation for adding additional evaluations on Vinoground and TOMATO to highlight the generalizability of the proposed approach to reasoning about real-world videos with subtle temporal cues. It is very promising to see using highly structured text templates for text-only training does not hurt the model's capability to understand more natural questions. After reading all the other reviews and authors' responses, I still think that this paper is insightful and will be a valuable contribution. Thus, I keep my initial rating. It would be great to include these additional experiments and discussions in the next version of the paper.

---

> > > ### Author Response · Authors · 2024-11-28
> > > **Thank you**
> > >
> > > Dear Reviewer hvnd,
> > >
> > > We sincerely appreciate your encouraging comments on our work. We have incorporated the additional results and discussion from the rebuttal, which we believe will strengthen our paper.
> > >
> > > Thank you once again for your positive support and valuable feedback!

---

### Official Review · Reviewer_UiVz · 2024-11-04

**Soundness:** 3
**Presentation:** 3
**Contribution:** 2
**Rating:** 6
**Confidence:** 4

**Summary:**

This paper tried to use text-only reordering style data to fine-tune MLLM to enhance the temporal ordering performance.

**Strengths:**

1. Show that visual features contain necessary information for order understanding task.
2. Writing is clear.

**Weaknesses:**

1. Why choosing LongVA? LongVA is only trained on images and text, therefore, evaluating and improving LongVA but evaluating on video understanding is not proper. Because everything is zero-shot manner. Instead, choosing backbones like LLaVA-onevision, phi-3.5-vision, or Qwen2-VL is proper. Those models have videos as the training data. If authors still show that there is clear improvement with their text-only ordering data, then the claim can be supported.
2. Probing visual features is unfair. For LLM (decoder only), authors did not fine-tune. For visual encoder features, authors fine-tuned. Since the task diversity is limited, therefore, overfitting to this fixed task is not hard. This also explains that a LSTM can overfit this task. The interesting thing should be: if you use LSTM after LLM features, can LSTM also overfits?
3. The empirical part of this paper demonstrate one thing: using text-only ordering data can improved relatively small LLM backbones. A more foundational question is: since large LLM can already solve the ordering task pretty well, is authors’ approach just a temporary  solution? Can authors’ approach improves Qwen-2-VL-72B?
4. Evaluation limited. Evaluations on benchmarks like NextQA, ActivityNet, etc is still necessary. This is because ordering task is naturally just a small subset of temporal reasoning.

**Questions:**

See comments above.

---

> ### Author Response · Authors · 2024-11-26
> **Response to Reviewer UiVz (Part 1)**
>
> Dear Reviewer UiVz,
>
> Thank you for your comments! Our responses to your questions are below.
>
> ### **W1-1: Why choose LongVA as the backbone?**
>
> We chose LongVA for two compelling reasons:
>
> 1. **Superior Performance and Rigorous Evaluation**
>    - LongVA demonstrates exceptional long video understanding capabilities (52.6 on Video-MME vs. 50.8 of Phi-3.5-vision). Using a strong baseline model allows us to rigorously validate our method's effectiveness. Besides, the shared Qwen-2 architecture between LongVA, Qwen2-VL, and LLaVA-OneVision ensures our findings are transferable across these models
>
> 2. **Consistency with our Claim: Temporal Reasoning Transfer without Video Data**
>    - LongVA's training exclusively on image-text data, with no video-language datasets. This setup is more consistent with our claim and provides a stronger validation of our findings.
>
> Regarding the alternative models (Qwen2-VL and LLaVA-OneVision) you mentioned, their recent release dates (Qwen2-VL: Sep. 19, LLaVA-OneVision: Sep. 25) were too close to the submission deadline (Oct. 1) for the comprehensive evaluation. Therefore we did not include the experiments with these models in our submitted version.
>
>
> ### **W1-2 and W3: Results on Larger Models and Video-trained Models**
>
> Thank you for raising this important question. Following your suggestion, we evaluated our method on Qwen2-VL-72B, a larger model pre-trained with video data.
> We used the ms-swift training framework following [Qwen2-VL fine-tuning best practices](https://swift.readthedocs.io/en/latest/Multi-Modal/qwen2-vl-best-practice.html).
> The evaluation results on MLVU and Video-MME using 16 and 32 frames demonstrate that T3 consistently improves performance even on large video-trained models:
>
>
> | MLVU                    | Action Count | Ego Reasoning (ER) | Needle Question-Answering (NQA) | Action Order | Plot Question-Answering (PQA) | Anomaly Recognition (AR) | Topic Reasoning (TR) | Macro-Average |
> | ----------------------- | ------------ | ------------------ | ------------------------------- | ------------ | ----------------------------- | ------------------------ | -------------------- | ------------- |
> | Qwen2-VL-72B (16 frame) | 20.4         | 52.3               | 55.8                            | 46.7         | 62.0                          | 65.5                     | 85.6                 | 55.5          |
> | + T3                    | 31.6 (+11.2) | 55.7 (+3.4)        | 67.0 (+11.2)                    | 54.1 (+7.4)  | 62.0 (-)                      | 61.0 (-4.5)              | 86.3 (+0.7)          | 59.7 (+4.2)   |
> | Qwen2-VL-72B (32 frame) | 22.8         | 54.8               | 65.1                            | 46.3         | 63.6                          | 71.0                     | 87.5                 | 58.7          |
> | + T3                    | 32.0 (+9.2)  | 59.9 (+5.1)        | 70.4 (+5.3)                     | 54.4 (+8.1)  | 64.0 (+0.4)                   | 69.0 （-2.0）            | 87.8 (+0.3)          | 62.5 (+3.8)   |
>
>
>
> | Video-MME                | Short       | Medium      | Long        | Overall     |
> | ------------------------ | ----------- | ----------- | ----------- | ----------- |
> | Qwen2-VL-72B  (16 frame) | 72.3        | 60.2        | 55.8        | 62.8        |
> | + T3                     | 73.8 (+1.5) | 62.6 (+2.4) | 54.9 (-0.9) | 63.7 (+0.9) |
> | Qwen2-VL-72B ( 32 frame) | 75.8        | 62.9        | 56.6        | 65.1        |
> | + T3                   | 78.3 (+2.5) | 64.3 (+1.4) | 57.1 (+0.5) | 66.6 (+1.5) |
>
> These results show that T3 provides significant improvements on key temporal reasoning tasks (e.g., Action Count: +9.2% and Action Order: +8.1% with 32 frames on MLVU) even when applied to a SOTA 72B video-trained model. This demonstrates that our approach is not merely a temporary solution for small models, but rather provides complementary benefits that enhance temporal understanding capabilities across model scales.

---

> > ### Author Response · Authors · 2024-11-26
> > **Response to Reviewer UiVz (Part 2)**
> >
> > ### **W2: Probing LLM Features**
> > This is an insightful question. We would like to address it from two perspectives:
> >
> > **Probing Visual Features**: Classifier probes are trained and tested on distinct sets of visual features to ensure validity. For instance, for *Order*, *Referring*, *Grounding* and *Blooming*, we collect training videos with a black background, while the testing videos all have a white background. This distinction ensures that **the probes on visual features do not overfit to the training set, but instead revealing the temporal information actually encoded in the visual features**.
> >
> > **Probing LLM Features**: Following your suggestion, we analyzed the LLM features by inputting videos into LongVA-7B and extracting the hidden states from the final layer of the LLM decoder. Two types of probes were then employed:
> > LSTM Probe: Trained on all hidden state features, this probe assesses whether temporal information is retained across all the final-layer LLM features.
> > Linear Probe: Trained on the hidden state of the last time step, this probe evaluates if the temporal information persists in a single hidden state.
> > The results are shown as follows. We can see that the LSTM probe's accuracy on LLM features is consistently lower than that on visual features across all temporal aspects, especially for the Attribute aspect (88.9% vs. 73.9%). The accuracy of the linear probe is even lower. These findings demonstrate that **a considerable portion of the temporal information is lost when transitioning from visual features to the final-layer LLM features**.
> >
> > Furthermore, we observe a substantial gap between the performance of the last-layer LLM features and the final output of LongVA. This suggests that even when relevant information is present in the LLM hidden states, the Video LLM can still produce incorrect answers. Investigating the causes of this discrepancy is left for future work.
> >
> > | Model                        | Attribute | Referring | Grounding | Order | Avg  |
> > |------------------------------|---------------|-----------|------------|-----------|------|
> > | LSTM Probe Visual Features   | **88.9**          | **92.7**      | **98.2**       | **96.4**      | **94.0** |
> > | LSTM Probe LLM Features      | 73.9          | 87.2      | 91.9       | 94.2      | 86.8 |
> > | Linear Probe LLM Features    | 72.7          | 80.0      | 80.4       | 80.4      | 78.4 |
> > | LongVA-7B                | 49.6          | 70.5      | 70.9       | 71.1      | 65.5 |

---

> ### Author Response · Authors · 2024-11-26
> **Response to Reviewer UiVz (Part 3)**
>
> ### **W4: Evaluation is limited**
> We respectfully disagree that our evaluation is limited. Beyond temporal ordering, our evaluations comprehensively cover various aspects of temporal reasoning through multiple established benchmarks:
>
> 1. MLVU and Video-MME are holistic video understanding benchmarks that evaluate diverse temporal reasoning capabilities including ego reasoning, action counting, plot understanding, and anomaly recognition.
>
> 2. We further validated our approach on two additional recent benchmarks with key results highlighted below (please refer to our responses for `Reviewer GraR` and `Reviewer hvnd`) :
>
> **TOMATO**: Tests sophisticated motion-related temporal reasoning:
> - Velocity & Frequency: T3 significantly improves performance (39.9% vs 23.1% baseline)
> - Shape & Trend: Shows notable gains (30.5% vs 22.4%)
> - These improvements demonstrate our method's generalization to unseen motion-related temporal concepts
>
> **Vinoground**: Evaluates fine-grained temporal understanding through real-world video-caption pairs with subtle temporal differences:
> - Text retrieval: +10.4% improvement (31.6% vs 21.2%)
> - Video retrieval: +8.4% improvement (30.0% vs 21.6%)
> - Group accuracy: +6.4% improvement (11.8% vs 5.4%)
>
> These results demonstrate that our approach generalizes well across different temporal reasoning aspects and datasets, beyond just ordering tasks. The improvements on tasks not explicitly present in T3 (e.g., Action Counting +3.9%, Plot QA +1.3% on MLVU) further validate the broad applicability of our method.
>
> Furthermore, while ActivityNet-QA and NextQA are valuable benchmarks, they may not be suitable for benchmarking the temporal reasoning capabilities according to [1]. Here's a detailed comparison:
>
>
> | Benchmark   | Avg. Duration (seconds) | Avg. Temporal Certificate Length (seconds)* | # Videos / # QA Pairs | Video Domains                                                |
> | ----------- | ------------------------ | ------------------------------------------- | --------------------- | ------------------------------------------------------------ |
> | ActivityNet | 180                      | 2.4                                         | 800 / 8000            | Everyday Life                                                |
> | NextQA      | 44                       | 2.7                                         | 1,000 / 8,564         | Everyday Life                                                |
> | Video-MME   | 1,018                    | 385.5                                       | 900 / 2,700           | Knowledge, Film & Television, Sports Competition, Life Record, and Multilingual |
> | MLVU        | 720                      | N / A                                       | 1,334 / 2,593         | Movies, TV Series, Egocentric Videos, Game                   |
>  **Temporal Certificate Length: the minimum video duration required for a human to verify the accuracy of an annotation [1]*
>
> Our chosen benchmarks are more suitable for evaluating temporal reasoning capabilities for several key reasons:
> 1. **Longer Temporal Dependencies**: Video-MME and MLVU feature significantly longer videos (1,018s and 720s vs 44-180s), allowing for evaluation of long-range temporal understanding.
>
> 2. **Temporal Verification Complexity**: The Temporal Certificate Length in Video-MME (385.5s) is substantially longer than ActivityNet (2.4s) and NextQA (2.7s), indicating that our chosen benchmarks require understanding broader temporal contexts rather than just short, isolated events.
>
> 3. **Domain Diversity**: While ActivityNet and NextQA focus solely on everyday activities, our benchmarks span multiple domains including knowledge-based content, entertainment, sports, and egocentric videos, offering a more comprehensive assessment of temporal reasoning abilities.
>
> These characteristics make our evaluation suite more appropriate for assessing sophisticated temporal understanding capabilities in real-world scenarios.
>
> [1] EgoSchema: A Diagnostic Benchmark for Very Long-form Video Language Understanding

---

> ### Author Response · Authors · 2024-11-29
> **Awaiting your response**
>
> Dear Reviewer UiVz,
>
> We have tried to address your concerns in our earlier response. If you have any further questions or suggestions, we are very happy to discuss with you.

---

> ### Author Response · Authors · 2024-12-02
> **Awaiting your response**
>
> Dear Reviewer UiVz,
>
> As we approach the end of the reviewer-author discussion period, we have made efforts to address your concerns in our previous response. We would greatly appreciate it if you could let us know whether our response has adequately addressed your questions and concerns.

---

### Official Review · Reviewer_GraR · 2024-11-04

**Soundness:** 2
**Presentation:** 2
**Contribution:** 2
**Rating:** 8
**Confidence:** 3

**Summary:**

This paper addresses the limitations of video LLMs in temporal reasoning, which involves tracking objects or events in a sequence of video frames. The authors discover and point out the deficiencies stemming from the limitations of LLM processing. Inspired by this, the authors present a text-based framework which can be used to finetune the LLM part of the video LLM, which can result in improved temporal understanding from text-only data. This approach enables smaller models to outperform larger, video-trained models on benchmarks such as TempCompass and Video-MME.

**Strengths:**

- Interesting approach to tackle temporal understanding of Video LLMs, that the LLM fails to understand the temporal behavior of the text prompts.
- Utilizing only text data proposes an efficient and scalable framework, although the gains are questionable for larger models.
- The performance gains are quite significant

**Weaknesses:**

- The biggest limitation of this work lies in the scope of the temporal concepts, but not simply because they were not covered in the paper. For example, consider rotation, direction, counting the number of a certain event, and relative velocity of an object relative to another object. There are some temporal concepts that cannot be represented in discrete sentences - for example, how would you describe the number of rotations of a diver in the form of text? How would you describe that one person is running faster than the other one without giving out the information explicitly?
- The above concern also brings up the question of whether the success of this approach is largely due to how video LLMs are tested on current video benchmarks: given a few discrete frames, the models are asked to answer a question. For example, a good video LLM would solve problems in "Order" when given a frame of book cover, die, school bus and then split-screen (example in figure 4). If the vision encoder encodes this information but the output quality is bottlenecked by the LLM, this method would improve the model. However, it is unclear whether this method will apply to questions that require virtually the entire video frames to be answered.

**Questions:**

- Are the temporal-oriented questions generated from a different distribution from the answer choice distribution of the video benchmarks? For example, are the orders in which the objects appear in the temporal-oriented questions not in the benchmark datasets? Does this still improve the performance when the training data is manipulated to be strictly out-of-distribution to the benchmark data?
- What is the performance of the fine-tuned video LLMs on the existing video benchmarks, when they are only provided the text part of the question?
- If there is an analysis of the number of data points provided and the performance gain, it could provide a better insights to the validity of this approach.

The first two questions were critical in my rating decision.

---

> ### Author Response · Authors · 2024-11-20
> **Response to Reviewer GraR (Part 1)**
>
> Dear Reviewer GraR,
>
> We greatly appreciate your thoughtful comments on our work. Below we address each concern in detail:
>
> ### **1. Can our method enhance the understanding of motion-related temporal concepts?**
> Thanks for raising this insightful question! Indeed, the temporal pattern of motions such as rotation, direction and velocity differ from concepts like order and temporal grounding, making them less straightforward to represent via frame-by-frame captioning. We acknowledge that fully understanding motion in videos may require approaches beyond our proposed textual temporal transfer. However, it is important to note that the temporal patterns of motion are inherently recognizable from discretely sampled video frames. For instance, by observing the relative position changes between two individuals across frames, we can infer which one is running faster. To interpret such information, a Video LLM must have a foundational understanding of temporal relationships between frames (e.g., order), a capability that can be supported by our method.
>
> To assess whether our method benefits the understanding of motion-related temporal concepts, we evaluate Video LLMs on the recently introduced TOMATO benchmark [1]. Interestingly, as shown in the table below, our method yields noticeable improvements in temporal concepts such as velocity & frequency (+16.8), shape & trend (recognizing the shape drawn by a person in the air) (+8.1) and action counting (+1.7). Notably, these temporal concepts are unseen in our training set. Table 2 and Table 3 in our paper also demonstrate that our T3 training data improves the accuracy of LongVA by 3.9 points in MLVU action counting and 2.4 points in TempCompass direction.
>
> We attribute these improvements to the enhanced ability to capture temporal relationship between video frames, which is the foundation for understanding velocity, direction, counting and shape & trend. These findings suggest that **our T3 training data, while primarily focused on basic temporal concepts, can generalize to more complex and unseen temporal concepts**. We will incorporate the TOMATO benchmark results and the above discussion into the next version of the paper.
>
> [1] TOMATO: Assessing Visual Temporal Reasoning Capabilities in Multimodal Foundation Models.
>
> **Performance on the TOMATO Benchmark.**
> | **Model**            | **Rotation** | **Direction** | **Velocity & Frequency** | **Shape & Trend** | **Visual Cues** | **Action Count** | **All** |
> | -------------------- | ------------ | ------------- | ------------------------ | ----------------- | --------------- | ---------------- | ------- |
> | GPT-4o               | 24.5         | 45.2          | 31.9                     | 42.6              | 58.6            | 36               | 37.7    |
> | LLaVA-NeXT-Video-32B | 20.6         | 26.3          | 12.4                     | 24.2              | 30              | 24.3             | 22.7    |
> | VideoLLaMA2 72B      | 14.3         | 24.6          | 22.4                     | 26.5              | 27.1            | 28.8             | 23.5    |
> | LongVA-7B            | 26.4         | 21.6          | 23.1                     | 22.4              | 38.6            | 14.7             | 22.3    |
> | + LLaVA-Next 200k    | 22.2         | 25.6          | 21.2                     | 26.0              | 40.0            | 19.8             | 21.2    |
> | + T3 (Ours) 200k     | 16.6         | 21.6          | 39.9                     | 30.5              | 38.6            | 16.4             | 24.3    |

---

> > ### Author Response · Authors · 2024-11-20
> > **Response to Reviewer GraR (Part 2)**
> >
> > ### **2. Is the distribution of our temporal-oriented questions different from the video benchmarks?**
> >
> > Thank you for highlighting this important question. To compare the distribution of questions between our T3 dataset and the video benchmarks, we conducted the following analysis:
> >
> > - **Data Sampling**: We utilized our validation set (each subset contains 500 examples) and randomly sampled 500 examples from the three video benchmarks.
> > - **Similarity Metric**: Question similarity was measured using Jaccard Similarity: $J(Q1, Q2) = \frac{|BoW(Q1) \cap BoW(Q2)|}{|Bow(Q1) \cup BoW(Q2)|}$. "BoW" refers to the bag-of-words representation of the questions, excluding stop words.
> > - **Similarity Analysis**: For each validation subset (e.g., Order, Referring, etc), we calculated the percentage of questions with a Jaccard Similarity greater than 0.5 compared to any question in the video benchmark.
> > - **Choice Distribution Comparison**: We conducted an analysis for answer choices in the same way, by treating each choice as an individual example.
> >
> > The results, presented below, show that only a small fraction of T3 dataset examples have similar counterparts in the video benchmarks. This indicates that our method's performance improvements are due to enhanced temporal reasoning capabilities, rather than simply memorizing the distribution of questions and answer choices.
> >
> > **Percentage of T3 validation set questions that are similar (>0.5 Jaccard Similarity) to any questions in the video benchmark. The results are averaged across five times of random sampling with the standard deviation reported in the ()**
> > |    |  TempCompass (%)  |   VideoMME (%)   | MLVU (%) |
> > |:------:|:-----:|:--------:|:--------:|
> > Attribute                  |  0.0(0.0) | 0.0(0.0) | 0.04(0.08) |
> > Order-Template (phrase)    |  0.0(0.0) | 0.0(0.0) |  0.0(0.0)  |
> > Order-Template (prefix)    |  0.0(0.0) | 0.0(0.0) |  0.0(0.0)  |
> > Order-Template (sentence)  |  0.0(0.0) | 0.0(0.0) |  0.0(0.0)  |
> > Order                      |  0.0(0.0) | 0.8(0.2) |  0.3(0.1)  |
> > Grounding                  |  0.0(0.0) | 0.0(0.0) |  0.0(0.0)  |
> > Referring                  |  0.0(0.0) | 0.0(0.0) | 0.24(0.20) |
> >
> > **Percentage of T3 validation set choices that are similar (>0.5 Jaccard Similarity) to any choices in the video benchmark.**
> > |    |  TempCompass (%)  |   VideoMME (%)   | MLVU (%) |
> > |:------:|:-----:|:--------:|:--------:|
> > Attribute                  |  1.5(0.4) | 0.1(0.1) |  2.7(0.5)  |
> > Order-Template (phrase)    |  0.0(0.0) | 0.0(0.0) |  0.0(0.0)  |
> > Order-Template (prefix)    |  0.0(0.0) | 0.0(0.0) |  0.0(0.0)  |
> > Order-Template (sentence)  |     -     |     -    |     -      |
> > Order                      |  0.7(0.4) | 0.8(0.2) |  0.3(0.1)  |
> > Grounding                  |  0.7(1.4) | 0.8(1.6) |  0.0(0.0)  |
> > Referring                  |  2.7(0.4) | 0.5(0.2) |  4.2(0.5)  |

---

> > > ### Author Response · Authors · 2024-11-20
> > > **Response to Reviewer GraR (Part 3)**
> > >
> > > ### **3. Performance of Video LLMs with Text-Only Input.**
> > >
> > > This is an interesting perspective. The table below reports the results on the three video benchmarks with text-only input, excluding the video. As we can see, LongVA-7B's performance drops significantly without visual input. Moreover, training with our T3 data does not yield any improvement under this scenario. This finding further supports the conclusion that our method's effectiveness is not dependent on memorizing the distribution of answer choices.
> > >
> > >
> > > | TempCompass                | Action | Direction | Speed | Order | Attribute_change | **Average** |
> > > | -------------------------- | ------ | --------- | ----- | ----- | ---------------- | ----------- |
> > > | LongVA-7B w/ video input   | 92.3   | 37.3      | 42.0  | 54.3  | 51.7             | **55.9**    |
> > > | LongVA-7B  w/o video input | 50.3   | 36.1      | 42.0  | 39.7  | 38.9             | 41.5        |
> > > | T3 (Ours) w/o video input  | 48.8   | 36.7      | 44.5  | 38.4  | 37.2             | 41.3        |
> > >
> > >
> > > | Video MME                 | Short | Medium | Long | Overall |
> > > | ------------------------- | ----- | ------ | ---- | ------- |
> > > | LongVA-7B w/ video input  | 61.1  | 50.4   | 46.2 | **52.6**    |
> > > | LongVA-7B w/o video input | 34.9  | 35.2   | 37.6 | 35.9    |
> > > | T3 (Ours) w/o video input | 34.8  | 35.3   | 34.4 | 34.9    |
> > >
> > >
> > > | MLVU                      | Action Count | Ego Reasoning (ER) | Needle Question-Answering (NQA) | Action Order | Plot Question-Answering (PQA) | Anomaly Recognition (AR) | Topic Reasoning (TR) | Macro-Average |
> > > | ------------------------- | ------------ | ------------------ | ------------------------------- | ------------ | ----------------------------- | ------------------------ | -------------------- | ------------- |
> > > | LongVA-7B w video input   | 25.2         | 48.6               | 70.4                            | 41.7         | 68.1                          | **58.5**                     | 82.2                 | 56.4          |
> > > | LongVA-7B w/o video input | 33.0         | 34.4               | 50.7                            | 36.3         | 43.2                          | 28.0                     | 36.7                 | 37.5          |
> > > | T3 (Ours) w/o video input | 32.5         | 34.1               | 49.3                            | 42.9         | 41.2                          | 25.5                     | 29.2                 | 36.4          |
> > >
> > > ### **4. Analysis of Training Data Quantity and Performance Gains.**
> > >
> > > Thank you for the constructive suggestion. As demonstrated in the table below, performance on Video-MME and MLVU improves consistently as we increase the amount of training data (0 -> 100K -> 200K). Additionally, Figure 9 in the appendix illustrates that textual validation accuracy also steadily rises with more training data. These results indicate that there is a positive correlation between textual and video temporal reasoning ability.
> > >
> > > | Model                   | Video-MME (S/M/L/Overall) | MLVU (Macro-Avg) |
> > > | ----------------------- | ------------------------- | ---------------- |
> > > | LongVA                  | 61.1 / 50.4 / 46.2/ 52.6  | 56.4%            |
> > > | + 100K T3 w/ LLaVA-Next | 61.7 / 51.0 / 45.7 / 52.8 | 56.9%            |
> > > | + 200K T3 w/ LLaVA-Next | 63.3 / 54.8 / 46.8 / 55.0 | 58.1%            |
> > >
> > > ### **5. Summary**
> > >
> > > Our additional analyses demonstrate that:
> > > 1. The method generalizes well to unseen temporal concepts (e.g. Velocity in the TOMATO benchmark) ;
> > > 2. Improvements come from enhanced temporal reasoning, not distribution memorization;
> > > 3. Performance scales consistently with training data.
> > >
> > > We hope these results could address the reviewer's concerns about the validity and scope of our approach. We will incorporate these results in our revision and discuss the TOMATO benchmark to highlight the generalization effect better.
> > >
> > > If you have further questions, we are happy to discuss :)

---

> > > > ### Comment · Reviewer_GraR · 2024-11-25
> > > > **Great Rebuttal**
> > > >
> > > > I thank the authors for their effort in providing detailed rebuttal. I have carefully read other reviews and the authors' rebuttals.
> > > >
> > > > Most of my concerns, especially the first two questions, were resolved. When I was reviewing the work, I was doubtful of whether their method will generalize to more complex tasks, and whether the train and test set would not overlap.
> > > >
> > > > In addition, I find the fact that their method does not yield much improvements in text-only setting very intriguing, and these facts noted during the rebuttal further solidifies their approach.
> > > >
> > > > Since those were the main concerns, I believe that if the authors include the discussion into their final manuscript, it would greatly strengthen the paper (and without the concerns I addressed, I think the methodology becomes much more interesting).
> > > >
> > > > Hence I decide to increase my score to 8.
> > > >
> > > > Thanks!

---

> > > > > ### Author Response · Authors · 2024-11-26
> > > > > **Thank you**
> > > > >
> > > > > We're glad that our rebuttal addressed your concerns and appreciate your recognition of our method.
> > > > >
> > > > > Your questions about generalization and methodology helped us better present our work's strengths and limitations.
> > > > > We will revise our paper to incorporate your valuable feedback to strengthen our manuscript.
> > > > >
> > > > > Thank you again for the timely response and efforts!

---

### Author Response · Authors · 2024-11-27
**General Response and Revision Summary**

We sincerely thank all reviewers for their constructive feedback.

We are encouraged that reviewers found our work addresses video-language reasoning from a novel and interesting perspective (hvnd, GraR), with clear writing and well-structured presentation (UiVz, SHtr). Reviewers highlighted several key strengths:
- (1) The insightful finding that LLM, rather than visual encoding, is the key bottleneck in video understanding (hvnd, c8nY).
- (2) The innovative and scalable approach of using text-only training to enhance temporal reasoning (hvnd, c8nY).
- (3) The extensive empirical analysis demonstrating significant performance gains (c8nY, GraR), even surpassing much larger models (SHtr).

During the rebuttal, we have substantially extended our work to address reviewers' concerns. We are particularly encouraged that Reviewers (GraR, hvnd, c8nY) found our additional results "solidify our approach" and "highlight its generalizability".
We accordingly include these valuable discussions in our revisions (highlighted in blue text for better visibility). The updates are sumarrized as follows:
- We incorporate the generalization evaluation on TOMATO & Vinoground in Table 5 and discuss the results in Section 4.3;
- We add new Appendix D to justify the performance gain, providing:
  1. distribution overlapping check between T3 and downstream tasks;
  2. model performance without video inputs;
  3. results on a large video-trained Video LLMs (Qwen2-VL-72B);
  4. The downstream correlation analysis is moved to Appendix D as well to meet the space constraint.
- Sec 4.1 with the highlighted motivation of choosing MLVU and Video-MME for evaluation;
- Appendix A.2 incorporates additional examples of frame captions;
- Appendix C.2 adds results of the dataset scaling effect on downstream tasks.

Thank you again for helping us strengthen our manuscript! We are happy to address any further questions.

---

### Meta-Review · Area_Chair_kaJv · 2024-12-21

**Metareview:**

This work presented an innovative perspective about the temporal reasoning ability for multimodal LLMs. The authors found that the drawback of temporal reasoning ability mainly stems from the weak performance of LLMs in this aspect. Starting from this observation, the authors curated a new text-only training data to teach the LLMs for temporal reasoning, and thus benefit the following multimodal temporal reasoning for video inputs. The results in the experiments and rebuttal session are solid and persuasive to demonstrate the effectiveness of the proposed method.

The main contribution of this work lies on the new insightful observations that the temporal reasoning capability of multimodal LLMs stems from that of LLMs. The authors solidly explained the methodology and conducted extensive experiments to demonstrate the effectiveness of the proposed method. After reading through the whole discussions between the authors and reviewers, the ACs think this is a great work to point out how to improve the highly-demanding capability in current multimodal LLMs for video understanding.

A few drawbacks were pointed out by the reviewers, and some of them are crucial. One of them is whether the LLMs do learn the temporal reasoning ability rather than memrozing the texts corpus. The authors did a great job to show that it is due to the latter reason for which we see the improvement of temporal reasoning capability. The auhors was initially challenged whether the proposed method can generalize to other benchmarks and LLM backbones. These were later also addressed well with solid experimental results. In the end, most of the concerns are addressed well, which lead to all positive ratings on this work.

To conclude, the ACs agree with the reviewers that this work shed some new lights on how to enhance the temporal reasoning capability for multimodal LLMs, which as we all know are crucial. Instead of approching this problem by constructing image-text instruction tuning data, the authors started from some intriguing observations and build a new method from the langauge perspective, whcih is quite insightful. As such, the ACs recommend an acceptance to this work.

**Additional Comments On Reviewer Discussion:**

The discussions between the authors and reviewers are engaged and enjoyable. It is a great exmaple of how those two groups should interact with each other. To the end, all reviewers reached to a good consensus and spoke highly of this work. As such, the ACs recommend a spotlight to this work and hope more audience could learn from this work.

---

### Decision · Program_Chairs · 2025-01-22

Accept (Poster)